# BiDRN: Binarized 3D Whole-body Human Mesh Recovery

## Abstract

3D whole-body human mesh recovery aims to reconstruct the 3D human body, face, and hands from a single image. Although powerful deep learning models have achieved accurate estimation in this task, they require enormous memory and computational resources. Consequently, these methods can hardly be deployed on resource-limited edge devices. In this work, we propose a Binarized Dual Residual Network (BiDRN), a novel quantization method designed to estimate the 3D human body, face, and hands parameters efficiently. Specifically, we design a basic unit Binarized Dual Residual Block (BiDRB) composed of Local Convolution Residual (LCR) and Block Residual (BR), which can preserve as much full-precision information as possible. For LCR, we further generalize it to four kinds of convolutional modules so that full-precision information can be propagated even across mismatched dimensions when reshaping features. Additionally, we also binarize the face and hands box-prediction network as Binarized BoxNet, which further reduces the model redundancy. Comprehensive quantitative and qualitative experiments demonstrate the effectiveness of BiDRN, which has a significant improvement over state-of-the-art binarization algorithms. Moreover, our BiDRN achieves comparable performance with the full-precision method Hand4Whole while using only **22.1%** parameters and **14.8%** operations. We will release all the code and pretrained models.

## 1 Introduction

3D whole-body human mesh recovery is a fundamental task in computer vision and aims to reconstruct the 3D whole-body human mesh of a person instance from a single image or video. By recovering the 3D whole-body human mesh, we are able to understand human behaviors and feelings through their poses and expressions. Therefore, 3D whole-body human mesh recovery has been widely

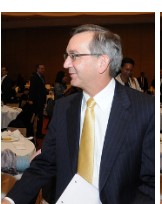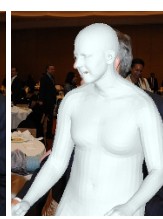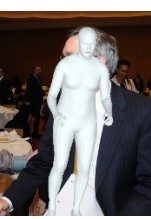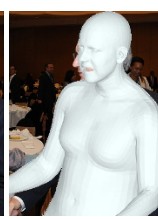

| | Image | Hand4Whole | BNN | BiDRN (Ours) |
|---|---|---|---|---|
| Params / OPs | | 77.84 / 16.85 | 21.61 / 2.63 | **17.22** / **2.50** |

Figure 1: Comparison of full-precision Hand4Whole, BNN, and our BiDRN. The second line is Parameters (M) / Operations (G).

applied for action recognition, virtual try-on, motion retargeting, and more. In recent years, powerful deep learning models (Choutas et al., 2020; Rong et al., 2021; Feng et al., 2021; Moon et al., 2022a; Lin et al., 2023) have been proposed with remarkable estimation accuracy. However, real-world applications like Augmented Reality (AR) require real-time responses, which necessitate the development of models that are accurate and efficient with less memory and computation cost.

Existing 3D whole-body human mesh recovery methods can be divided into two categories, *i.e.,* optimization-based methods and regression-based methods. The latter is more efficient and gains more attention with the rise of SMPL (Loper et al., 2023) and SMPL-X (Pavlakos et al., 2019) parametric models. Most regression-based models (Choutas et al., 2020; Rong et al., 2021; Feng et al., 2021; Moon et al., 2022a; Zhou et al., 2021) contain separate body, hands, and face networks. Hands and face regions are cropped from the original image with predicted boxes. Then, they are resized into higher resolution and input to the hands and face encoders respectively to achieve better estimation. The encoder of each network extracts image features, whose quality is required,

and feeds them into the decoder for regressing the corresponding body, hands, and face parameters. Finally, these parameters are fed into an SMPL-X layer (Pavlakos et al., 2019) to obtain a 3D whole-body human mesh. Although superior performance is achieved, they usually have a large model size and require extensive computing and memory resources, especially high-end GPUs. In addition, methods like Hand4Whole (Moon et al., 2022a) adopt a multi-stage pipeline with additional hand-only and face-only datasets (Moon et al., 2020; Zimmermann et al., 2019), which results in a more complicated system. The demand for running 3D whole-body human mesh recovery on mobile devices (with limited resources) increases rapidly. It is urgent to develop a simple yet efficient algorithm for 3D reconstruction while preserving the estimation accuracy as much as possible.

As deep learning models grow rapidly in size, model compression becomes crucial, particularly for deployment on edge devices. Relevant research can be divided into five categories, including quantization (Xia et al., 2023; Qin et al., 2020b;a; Hubara et al., 2016; Zhou et al., 2016; Liu et al., 2018), knowledge distillation (Hinton et al., 2015; Chen et al., 2018; Zagoruyko & Komodakis, 2017), pruning (Han et al., 2015; 2016; He et al., 2017), lightweight network design (Howard et al., 2017; Zhang et al., 2018; Ma et al., 2018), and low-rank approximation (Denton et al., 2014; Lebedev et al., 2015; Lebedev & Lempitsky, 2016). Among these, binarized neural network (BNN) is the most aggressive quantization technology that can compress memory and computational costs extremely. By quantizing the full-precision (32 bits) weights and activations into only 1 bit, BNN achieves significant computational efficiency, offering up to $32\times$ memory saving and $58\times$ speedup on CPUs for convolution layer (Rastegari et al., 2016). Additionally, bitwise operations like XNOR can be efficiently implemented on embedded devices (Zhang et al., 2019; Ding et al., 2019).

However, the direct application of network binarization for 3D whole-body human mesh recovery may encounter three challenges: **(1)** The quality of extracted features from the encoder is significant for parameter regression. Directly binarizing the encoder may cause severe full-precision information loss. **(2)** The dimension mismatch problem, when reshaping features, prevents bypassing full-precision information in BNN, which should be tackled for general situations. **(3)** To obtain accurate enough body, hands, and face parameters with as little memory and computation cost as possible, which parts should or should not be binarized requires careful consideration.

To address the above challenges, we propose **Bi**narized **D**ual **R**esidual **N**etwork (BiDRN), a novel BNN-based methods for 3D whole-body human mesh recovery. **First**, we propose a Binarized Dual Residual Block (BiDRB), which serves as a basic unit of the network. Specifically, BiDRB can bypass full-precision activations, which is significant for body, hands, and face parameter regression, by adopting a Local Convolution Residual (LCR) with almost the same memory and computation cost. Besides, we redesign four kinds of convolutional modules and generalize them to more complicated situations so they can apply the LCR even for dimension mismatch situations. Moreover, BiDRB utilizes a full-precision Block Residual (BR) to further enhance the full-precision information with tolerable cost but significant improvements. **Second**, we binarized specific layers in the hands and face box-prediction net, which can maintain the performance while significantly reducing memory and computation costs. Based on these techniques, we derive our BiDRN that significantly improves over SOTA BNNs, with more than 31.5 *All MPVPEs* reduction, as shown in Figure 2.

Our contributions can be summarized as follows.

- We propose BiDRN, a novel BNN-based method for the task of 3D whole-body human mesh recovery. To the best of our knowledge, this is the first work to study the binarization of the 3D whole-body human mesh recovery problem.
- We propose a new binarized unit BiDRB composed of Local Convolution Residual (LCR) and Block Residual (BR), which can maintain the full-precision information as much as possible and narrow the *All MPVPEs* gap with the full-precision method from **85.9** to **32.0**.
- Our BiDRN not only significantly outperforms existing SOTA BNNs, but it also achieves even comparable performance with the full-precision Hand4Whole method while requiring less than a quarter of the parameters and calculations.

## 2 RELATED WORK

**Whole-body Human Mesh Recovery.** Optimization-based methods (Joo et al., 2018; Xiang et al., 2019; Pavlakos et al., 2019; Xu et al., 2020) first estimate 2D person keypoints, and then reconstruct

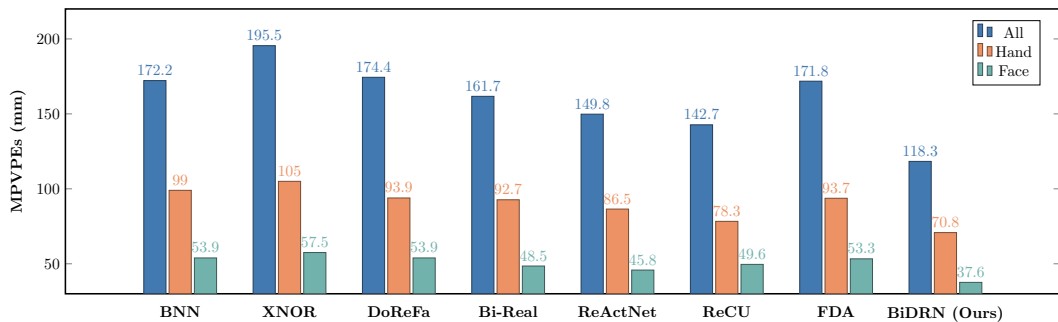

Figure 2: Comparison between recent BNNs and our BiDRN on EHF dataset. *MPVPEs* (the lower, the better) of All, Hand, and Face are depicted in blue, orange, and green respectively. BiDRN significantly reduces the *All MPVPEs* of BNN (Hubara et al., 2016), XNOR (Rastegari et al., 2016), DoReFa (Zhou et al., 2016), Bi-Real (Liu et al., 2018), ReActNet (Liu et al., 2020), ReCU (Xu et al., 2021b) and FDA (Xu et al., 2021a) by 53.9, 77.2, 56.1, 43.4, 31.5, 24.4, and 53.5 respectively.

3D human bodies with additional constraints. Yet, these methods often involve complex optimization objectives and thus are computationally intensive. With the release of statistical human models, like SMPL (Loper et al., 2023) and SMPL-X (Pavlakos et al., 2019), regression-based methods emerge to recover the 3D human mesh in an end-to-end manner. For example, ExPose (Choutas et al., 2020) utilizes body-driven attention to extract crops of face and hand regions and part-specific knowledge from existing face- and hand-only datasets. FrankMocap (Rong et al., 2021) first runs 3D pose regression methods for body, face, and hands independently, followed by composing the regression outputs via an integration module. PIXIE (Feng et al., 2021) proposes a novel moderator to fuse body part features adaptively with realistic facial details. Hand4Whole (Moon et al., 2022a) produces more accurate 3D wrist rotation and smooth connection between 3D whole-body and hands by combining both body and hand MCP joint features. Although these powerful methods achieve precise results of 3D human mesh, they require powerful hardware with enormous memory and computation resources. Moreover, they utilize multi-stage pipelines for body, hands, and face estimation, which further increases the training difficulty and resource consumption. 3D whole-body human mesh recovery models that can store and run in resource-limit devices are being under-explored. This work tries to move forward in this direction.

**Binarized Neural Network.** Binarized neural networks (BNNs) (Hubara et al., 2016) represent both the activations and weights with only 1-bit, providing an extreme level of compression for computation and memory. It is first introduced in the image classification task, and several follow-up works (e.g., Bi-Real (Liu et al., 2018), ReActNet (Liu et al., 2020), and IR-Net (Qin et al., 2020b)) further push the performance boundary, making substantial improvements over the original implementation. Due to BNN's ability to achieve extreme model compression while delivering relatively acceptable performance, it has also been widely applied in other vision tasks. For example, Jiang et al. (2021) proposes a BNN without batch normalization for image super-resolution task. Cai et al. (2023) designed a binarized convolution unit BiSR-Conv that can adapt the density and distribution of hyperspectral image (HSI) representations for HSI restoration. However, the potential of BNN in 3D whole-body human mesh recovery remains unexplored.

## 3 METHOD

Considering that a line of outstanding 3D reconstruction works is based on the ResNet backbone, we propose our binarization model based on the SOTA ResNet-based method Hand4Whole (Moon et al., 2022a). Hoping that our binarization method can benefit these works and provide a fair comparison. In Hand4Whole, ResNet (He et al., 2016) backbone plays a pivotal role in extracting detailed and high-quality features from the body, face, and hands, which is the main source of memory and computational costs. In addition, it uses the extracted body feature to predict the bounding box of face and hands by a BoxNet, which may be complex and can be compressed as well. Based on these observations, we propose a Binarized Dual Residual Network (BiDRN) (see Figure 3) to replace the ResNet backbone and a Binarized BoxNet. They can reduce memory and computational costs enormously while preserving accuracy.

Figure 3: The overview pipeline of our binarized 3D whole-body human mesh recovery method. The body, hand, and face BiDRN serve as encoders to extract corresponding features. Binarized BoxNet predicts the face and hand regions based on the body features.

## 3.1 BINARIZED DUAL RESIDUAL BLOCK

The details of BiDRB are illustrated in Figure 4. The full-precision activation input $\boldsymbol{a}_f \in \mathbb{R}^{C \times H \times W}$ is binarized into 1-bit activation by a Sign function as

$$a_b = \text{Sign}(a_f) = \begin{cases} +1, & a_f \geq 0 \\ -1, & a_f < 0 \end{cases}, \tag{1}$$

where $\boldsymbol{a}_b \in \mathbb{R}^{C \times H \times W}$ denotes the binarized activation. Yet, the Sign function is non-differentiable and we have to approximate it during backpropagation. Here, we adopt a piecewise quadratic function to smoothly approximate the Sign function during the gradient computation process as

$$F(a_f) = \begin{cases} +1, & a_f \geq 1 \\ -a_f^2 + 2a_f, & 0 \leq a_f < 1 \\ a_f^2 + 2a_f, & -1 \leq a_f < 0 \\ -1, & a_f < -1 \end{cases}. \tag{2}$$

We find the ReLU pre-activation used by default in previous work will generate all-one activations after the Sign function. This may lead to the failure of binarization. To solve it, we adopt a Hardtanh pre-activation function that can compress the full-precision activation into the range $[-1, +1]$ as

$$a_f = \text{Hardtanh}(x_f) = \begin{cases} +1, & x_f \geq 1 \\ x_f, & -1 \leq x_f < 1 \\ -1, & x_f < -1 \end{cases}, \tag{3}$$

where $\mathbf{X}_f \in \mathbb{R}^{C \times H \times W}$ represents the output feature map generated by the preceding layer. Compared with methods that use a learnable threshold before the Sign function (Liu et al., 2020) or applying a redistribution trick (Cai et al., 2023), the Hardtanh pre-activation can achieve better performance without introducing additional parameter burden.

Quantizing weights by the same Sign function can extremely reduce the parameters, thus weights $\mathbf{W}_f \in \mathbb{R}^{C_{\text{in}} \times C_{\text{out}} \times K \times K}$ in binarized convolution layer is quantized into scaled 1-bit weights $\mathbf{W}_b$ as

$$w_b^i = \alpha^i \cdot \text{Sign}(w_f^i), \tag{4}$$

where index $i$ represents the $i$-th output channel, and $\alpha^i$ is a scaling factor defined as $\alpha^i = \frac{\|w_f^i\|_1}{C_{\text{in}} \times K \times K}$. Multiplying the binarized weights by channel-wise scaling factor can better maintain the original distribution of weights on each channel. After binarizing both the activations and weights, the computation of binarized convolution can be simply formulated as (Rastegari et al., 2016)

$$\boldsymbol{o} = \alpha \cdot \text{bitcount}(\text{Sign}(\boldsymbol{a}_f) \odot \text{Sign}(\mathbf{W}_f)), \tag{5}$$

where $\odot$ denotes the XNOR-bitcount bitwise operation between binarized activations and weights, and $\boldsymbol{o}$ denotes the output of binarized convolution.

XNOR and bitcount are both logical operations that can significantly reduce the computation overhead of full-precision matrix multiplication. However, the loss of full-precision information in quantization is non-neglectable. Compared with the binarized information, full-precision information

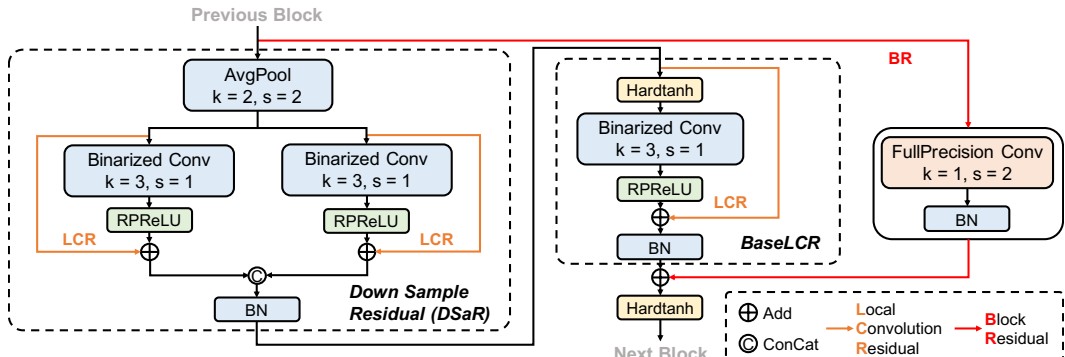

Figure 4: A kind of Binarized Dual Residual Block (BiDRB). The orange arrow denotes Local Convolution Residual (LCR), while the red arrow denotes Block Residual (BR).

usually represents image details, which may not be dominant in the image classification task, but is significant in 3D body mesh recovery. Since regression-based methods only optimize a few body, hands, and face parameters, even slight perturbations on the feature space may be transmitted to the parameters and have a great impact on the final 3D human mesh.

To preserve the full-precision information as much as possible, we design two kinds of residual connections, *i.e.,* Local Convolution Residual (LCR) and Block Residual (BR) as follows.

**Local Convolution Residual.** This residual connection is applied to each binarized convolution layer to bypass full-precision activation. Since the value range of binarized output $o$ is much smaller than that of full-precision activation $a_f$, we first apply the channel-wise RPReLU (Liu et al., 2020) activation function to enlarge its value diversity and redistribute the representation as

$$\text{RPReLU}(o^i) = \begin{cases} o^i - \gamma^i + \zeta^i, & o^i > \gamma^i \\ \beta^i(o^i - \gamma^i) + \zeta^i, & o^i \leq \gamma^i \end{cases},$$ (6)

where $o^i$ is the binarized convolution output of the $i$-th channel, $\gamma^i, \zeta^i, \beta^i \in \mathbb{R}$ are learnable parameters. After that, the full-precision activation $a_f$ is added as

$$o' = \text{BatchNorm}(\text{RPReLU}(o) + a_f),$$ (7)

where $o'$ is the output feature. Note that the parameters introduced by RPReLU are relatively small compared to the convolution kernels and thus can be ignored.

This local convolution residual can bypass full-precision information during the whole network if the dimension remains unchanged. Unfortunately, to extract compact image features, there exists Down Scale, Down Sample, Fusion Up, and Fusion Down operations in the encoder. The dimension mismatch problem in these modules prevents bypassing the full-precision information and thus leads to a performance drop. To tackle this problem, we redesign these modules so that they can be combined with our Local Convolution Residual, as illustrated in Figure 5.

Specifically, Down Scale module reduces the spatial dimension of the input feature map. To match the output dimension, the full-precision activation is first fed into an average pooling function and then added to the output of Down Scale convolution as

$$o' = \text{BatchNorm}(\text{RPReLU}(o) + \text{AvgPool}(a_f)),$$ (8)

where $o', o \in \mathbb{R}^{C \times \frac{H}{2} \times \frac{W}{2}}, a_f \in \mathbb{R}^{C \times H \times W}$. Average pooling does not introduce any additional parameter and its computational cost can be ignored compared to the encoder.

For Fusion Up which increases the channel dimension, we replace the single convolution layer with two distinct layers. The design is guided by the principle of maintaining the output channel count of each layer equivalent to its input channel count. By aligning the input and output dimensions in this manner, the layers can seamlessly integrate with the normal Local Convolution Residual (LCR) mechanism, which helps in retaining the original full-precision information. Finally, the outputs of these two layers are concatenated in channel dimension as

$$o' = \text{BatchNorm}(\text{Concat}(o'_1, o'_2)),$$ (9)

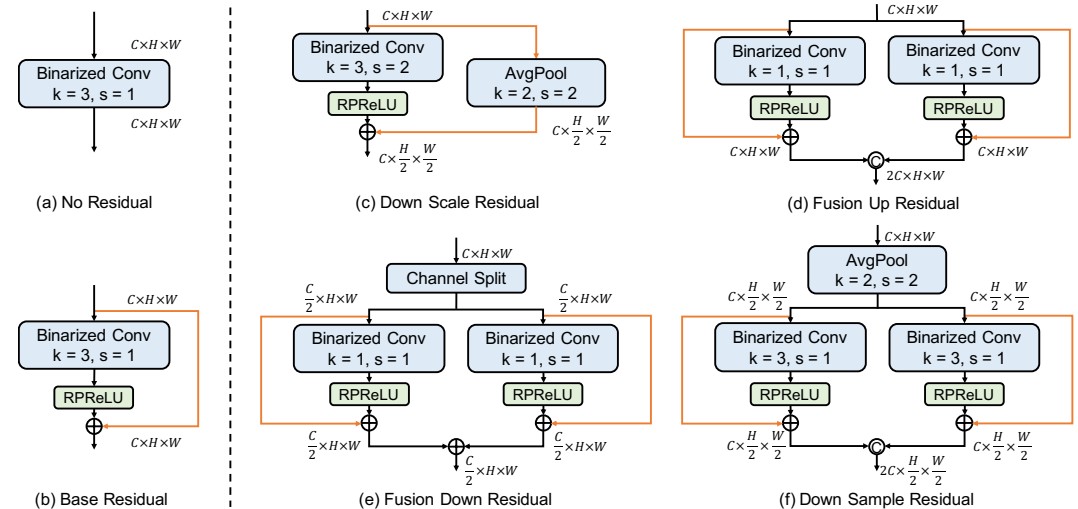

Figure 5: Illustration of our Local Convolution (Base) Residual and four redesign modules, including (c) Down Scale Residual (DScR), (d) Fusion Up Residual (FUR), (e) Fusion Down Residual (FDR), and (f) Down Sample Residual (DSaR). The orange arrow denotes the full-precision information flow. For simplicity, batch normalization and Hardtanh pre-activation are omitted.

where $\boldsymbol{o}' \in \mathbb{R}^{2C \times H \times W}, \boldsymbol{o}'_1, \boldsymbol{o}'_2 \in \mathbb{R}^{C \times H \times W}$.

Fusion Down is the inverse of Fusion Up, thus we first split the input w.r.t. channel and then feed them into two distinct binarized convolution layers with LCR. Finally, they are summed up as

$$\boldsymbol{o}' = \text{BatchNorm}(\boldsymbol{o}'_1 + \boldsymbol{o}'_2), \tag{10}$$

where $\boldsymbol{o}' \in \mathbb{R}^{\frac{C}{2} \times H \times W}, \boldsymbol{o}'_1, \boldsymbol{o}'_2 \in \mathbb{R}^{\frac{C}{2} \times H \times W}$.

Down Sample is the combination of both Down Scale and Fusion Up, where we first apply the average pooling and then employ the channel concatenation. Note that we just describe the condition of double or half the size for simplicity, while it is generalized to more complex conditions with four times channels in BiDRN (see supplementary file). By redesigning these four modules, we are able to bypass the full-precision activations with almost the same parameter and computational cost.

**Block Residual.** Full-precision information may be diluted by binarized convolution layers, particularly in very deep networks. To address this, we propose a Block Residual mechanism that bypasses full-precision information in each block, preserving crucial details throughout the network.

Note that the number of blocks is significantly lower than the count of convolution layers, we utilize a full-precision Conv1×1 layer to extract more accurate features with acceptable parameter burden. As shown in Figure 4, the overall BiDRB composed of both LCR and BR can be formulated as

$$\boldsymbol{o}'' = \text{BaseLCR}(\text{DSaR}(a_f)) + \text{BR}(a_f), \tag{11}$$

where BaseLCR, DSaR, and BR denote base Local Convolution Residual, Down Sample Residual, and Block Residual respectively. Note that Equation (11) is only one kind of BiDRB, other kinds of BiDRB may incorporate alternative modules such as Fusion up, Fusion Down, and Down Scale Residuals. Moreover, it is worth noting that a binarized version of Block Residual can be used for tasks that do not require high-quality features but require efficiency with extreme compression.

By combining both Local Convolution Residual and Block Residual, Binarized Dual Residual Block can preserve full-precision information as much as possible while maintaining nearly the same number of parameters and computational cost. The body, hand, and face encoders that build on BiDRN can extract better image features than simple binarization methods.

## 3.2 BINARIZED BOXNET

The bounding boxes for the hands and face are predicted by BoxNet. Initially, it predicts 3D heatmaps of human joints $\mathbf{H}$ from the encoder output $\mathbf{F}$. These heatmaps are then concatenated with $\mathbf{F}$, and several Deconv and Conv layers are applied to this combined feature map. Afterward,

Table 1: 3D whole-body reconstruction error comparisons on EHF (Pavlakos et al., 2019) and AGORA (Patel et al., 2021) benchmarks. † indicates that the model does not use pre-trained weights, as well as additional hand-only and face-only datasets for fair comparison.

| Method | Bit | Params (M) | OPs (G) | EHF | | | | | | | | AGORA | | | | | |
| | | | | MPVPE ↓ | | | PA-MPVPE ↓ | | | PA-MPJPE ↓ | | MPVPE ↓ | | | PA-MPVPE ↓ | | |
| | | | | All | Hand | Face | All | Hand | Face | Body | Hand | All | Hand | Face | All | Hand | Face |
|---|---|---|---|---|---|---|---|---|---|---|---|---|---|---|---|---|---|
| ExPose | 32 | - | - | 77.1 | 51.6 | 35.0 | 54.5 | 12.8 | 5.8 | - | - | 219.8 | 115.4 | 103.5 | 88.0 | 12.1 | 4.8 |
| FrankMocap | 32 | - | - | 107.6 | 42.8 | - | 57.5 | 12.6 | - | - | - | 218.0 | 95.2 | 105.4 | 90.6 | 11.2 | 4.9 |
| PIXIE | 32 | - | - | 89.2 | 42.8 | 32.7 | 55.0 | 11.1 | 4.6 | - | - | 203.0 | 89.9 | 95.4 | 82.7 | 12.8 | 5.4 |
| Hand4Whole † | 32 | 77.84 | 16.85 | 86.3 | 47.2 | 26.1 | 57.5 | 13.2 | 5.8 | 70.9 | 13.3 | 194.8 | 78.6 | 88.3 | 79.0 | 9.8 | 4.8 |
| BNN | 1 | 21.61 | 2.63 | 172.2 | 99.0 | 53.9 | 115.6 | 18.4 | 6.2 | 129.4 | 19.0 | 267.6 | 114.0 | 141.3 | 94.9 | 10.4 | 5.0 |
| XNOR | 1 | 21.61 | 2.63 | 195.5 | 105.0 | 57.5 | 119.9 | 18.5 | 6.2 | 134.5 | 19.1 | 271.1 | 127.9 | 156.9 | 94.1 | 10.5 | 5.1 |
| DoReFa | 1 | 21.61 | 2.63 | 174.4 | 93.9 | 53.9 | 109.3 | 18.4 | 6.0 | 121.3 | 19.0 | 257.6 | 115.3 | 139.4 | 93.5 | 10.4 | 5.0 |
| Bi-Real | 1 | 21.61 | 2.63 | 161.7 | 92.7 | 48.5 | 108.7 | 18.5 | 5.9 | 121.2 | 19.1 | 242.0 | 104.3 | 121.8 | 92.6 | 10.4 | 5.0 |
| ReActNet | 1 | 21.66 | 2.63 | 149.8 | 86.5 | 45.8 | 98.8 | 18.5 | 6.1 | 111.6 | 19.1 | 237.6 | 102.9 | 120.2 | 91.4 | 10.4 | 4.9 |
| ReCU | 1 | 21.71 | 2.65 | 142.7 | 78.3 | 49.6 | 85.4 | 18.2 | 6.0 | 97.1 | 18.8 | 225.1 | 96.2 | 108.3 | 89.7 | 10.3 | 4.9 |
| FDA | 1 | 32.06 | 2.81 | 171.8 | 93.7 | 53.3 | 108.5 | 18.4 | 6.1 | 120.5 | 19.0 | 256.4 | 114.6 | 138.6 | 93.0 | 10.4 | 5.0 |
| **BiDRN (Ours)** | 1 | 17.22 | 2.50 | **118.3** | **70.8** | **37.6** | **76.9** | **17.4** | **6.0** | **88.2** | **17.9** | **215.0** | **92.1** | **102.3** | **87.7** | **10.3** | **4.9** |

soft-argmax (Sun et al., 2018) is used to determine the box center, followed by fully connected layers to compute the box size. We observe that the parameters and computational cost of these layers, especially the Deconv layers, are significantly higher compared to other components in the decoder, which seems excessive for calculating a few bounding box parameters.

Thus, we binarize both Deconv layers and Linear layers except the final one in Figure 6, so that we can maintain good output accuracy. Experiments (Table 3) further show that such binarization even leads to performance gain and meanwhile reduces memory and computational costs significantly.

**Loss Function.** By combining BiDRB with the binarized BoxNet, we obtain our final model BiDRN. Following (Moon et al., 2022a), we train it end-to-end by minimizing the following loss:

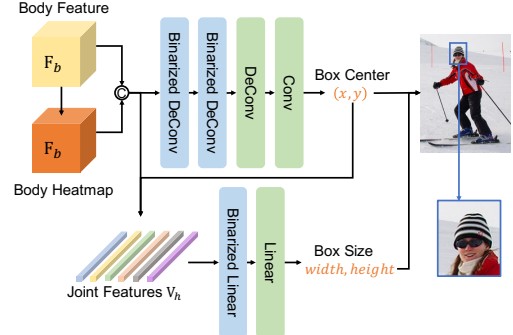

$$L = L_{smplx} + L_{joint} + L_{box}, \qquad (12)$$

where $L_{smplx}$ is the $L1$ distance between predicted and GT SMPL-X parameters, $L_{joint}$ is the $L1$ distance between predicted and GT joint coordinates, and $L_{box}$ is the L1 distance between predicted and GT hands and face bounding boxes.

Figure 6: Binarized face BoxNet extracts the face region from the high-resolution human image. Hand regions are extracted by binarized hands BoxNet with the same architecture.

## 4 EXPERIMENT

### 4.1 EXPERIMENTAL SETTINGS

**Datasets.** We use Human3.6M (Ionescu et al., 2014), whole-body MSCOCO (Jin et al., 2020) and MPII (Andriluka et al., 2014) for training. Following Moon et al. (2022a), the 3D pseudo-GTs for training are obtained by NeuralAnnot (Moon et al., 2022b). To make the binarized model simple and easy to train, different from Moon et al. (2022a), we do not use additional hand-only and face-only datasets, or additional stages to finetune the model. Finally, we evaluate our BiDRN on EHF (Pavlakos et al., 2019) and AGORA (Patel et al., 2021).

**Evaluation Metrics.** We adopt Mean Per Joint Position Error (MPJPE) and Mean Per Vertex Position Error (MPVPE), along with their aligned version PA-MPJPE and PA-MPVPE, to evaluate the performance of 3D whole-body human mesh recovery. Consistent with prior works (Xia et al., 2023; Qin et al., 2020b; Hubara et al., 2016; Cai et al., 2023), we calculate the parameters of BNN-based methods as Params = Params$_b$ + Params$_f$, where Params$_b$ = Params$_f$ / 32 represents that the binarized parameters is 1/32 of its full-precision counterpart. Similarly, the computational complexity of BNNs is measured by operation per second (OPs), which is calculated as OPs = OPs$_b$ + OPs$_f$, where OPs$_b$ = OPs$_f$ / 64, and OPs$_f$ = FLOPs (floating point operations).

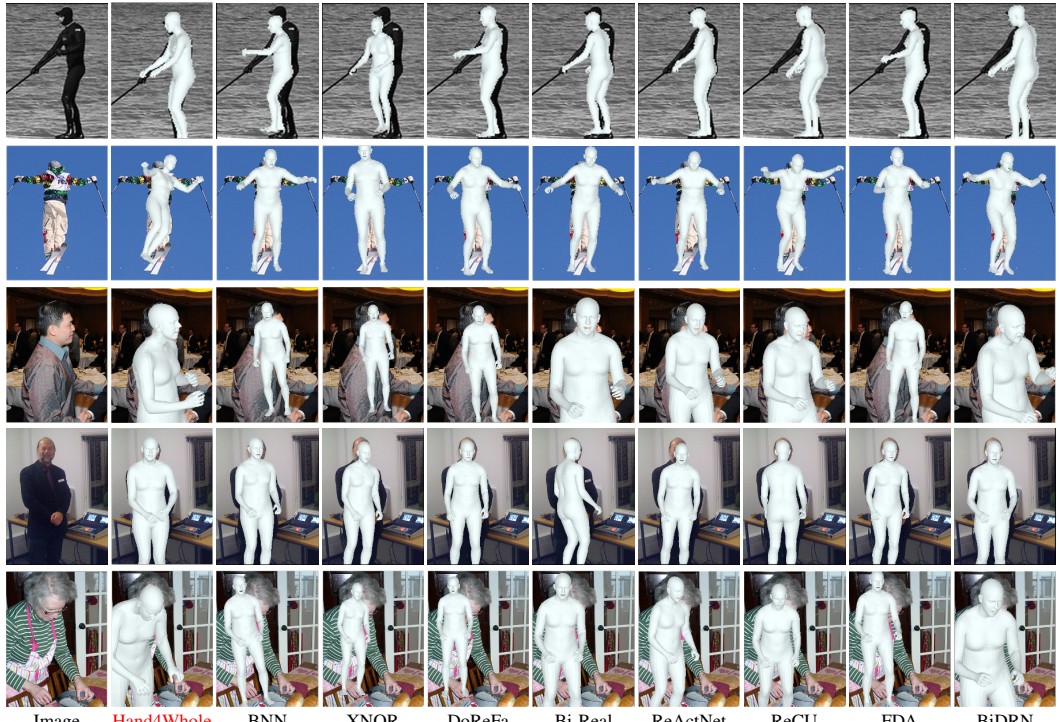

Figure 7: Qualitative comparison between full-precision Hand4Whole, seven existing SOTA BNN-based methods and our newly proposed BiDRN on the MSCOCO (Jin et al., 2020) dataset. Bypassing the full-precision information is necessary for accurate whole-body human mesh recovery.

**Implementation Details.**  Our BiDRN is implemented in PyTorch (Paszke et al., 2019). To make the whole pipeline more concise, and more importantly, validate that the great performance of our BiDRN is not due to a large pretraining dataset or some finetune tricks, we do not pre-train it on any dataset, nor finetune by additional hand-only and face-only datasets. We use Adam (Kingma & Ba, 2015) optimizer with batch size 24 and initial learning rate of $1 \times 10^{-4}$ to train BiDRN for 14 epochs on a single A100-80G GPU. We apply standard data augmentation techniques, including scaling, rotation, random horizontal flipping, and color jittering. We also provide a mapping table from ResNet backbones to the proposed modules of BiDRN in supplementary file.

### 4.2 QUANTITATIVE RESULTS

We compare BiDRN with 7 SOTA BNN-based methods: BNN (Hubara et al., 2016), XNOR (Rastegari et al., 2016), DoReFa (Zhou et al., 2016), Bi-Real (Liu et al., 2018), ReActNet (Liu et al., 2020), ReCU (Xu et al., 2021b), and FDA (Xu et al., 2021a). To adapt these general-purpose BNNs to the reconstruction task, we replace the convolutional layers in Hand4Whole's ResNet backbone with binary convolutions from the corresponding BNN. The rest of the model remains unchanged, following the convention for model binarization. Besides, we also compare it with 4 SOTA 32-bit full-precision methods, including ExPose (Choutas et al., 2020), FrankMocap (Rong et al., 2021), PIXIE (Feng et al., 2021), and Hand4Whole (Moon et al., 2022a).

Table 1 presents the performance comparisons on both EHF and AGORA datasets. It can be observed that although existing SOTA BNN-based methods can compress the model to only 27.8% (21.61/77.84) of the original Params and 15.6% (2.63/16.85) of the original OPs, directly applying them to the 3D mesh recovery task achieves poor performance. In comparison, our BiDRN achieves superior performance compared to these SOTA BNN-based methods with even fewer parameters and operations demands. Specifically, the *All MPVPEs* of BiDRN show 31.3%, 39.5%, 32.2%, 26.8%, 21.0%, 17.1%, and 31.1% improvements than BNN, XNOR, DoReFa, Bi-Real, ReActNet, ReCU, and FDA on EHF dataset respectively. Furthermore, the AGORA dataset results reinforce the strengths of BiDRN. As shown in the right half of Table 1, BiDRN continues to outperform 7 SOTA BNN-based methods. Compared to the most basic BNN algorithm, the *MPVPEs* of our BiDRN show 19.7%, 19.2%, and 27.6% improvements on body, hands, and face respectively.

Table 2: Ablation study on the EHF dataset. All experiments are evaluated using *MPVPEs*, with final results highlighted in **bold**. In table (a), DScR, FUR, FDR, and DSaR denote the Down Scale Residual, Fusion Up Residual, Fusion Down Residual, and Down Sample Residual of Figure 5 respectively. In table (d), the *MPVPEs* of binarizing all networks are 118.3, 70.8, 37.6 for All, Hand, and Face respectively, while the *MPVPEs* of full-precision network are 86.3, 47.2, 26.1 respectively.

(a) Break-down ablation of LCR

| Method | BaseLCR | + DScR | + FUR | + FDR | + DSaR |
|---|---|---|---|---|---|
| All MPVPEs | 139.3 | 127.8 | 126.0 | 124.7 | **118.3** |
| Params (M) | 17.05 | 17.05 | 17.14 | 17.21 | 17.22 |
| OPs (G) | 2.48 | 2.48 | 2.49 | 2.50 | 2.50 |

(b) Study of pre-activation function

| Method | Additional Params | All | Hand | Face |
|---|---|---|---|---|
| Hardtanh($x_f$) | No | **118.3** | **70.8** | **37.6** |
| ReLU($x_f$) | No | 126.8 | 71.5 | 38.9 |
| PReLU($x_f$) | Yes | 125.9 | 70.6 | 37.3 |

(c) Ablation study of Block Residual (BR)

| Method | Params (M) | OPs (G) | All | Hand | Face |
|---|---|---|---|---|---|
| w/o BR | 11.51 | 1.25 | 139.6 | 85.4 | 39.1 |
| Binarized BR | 11.68 | 1.28 | 120.0 | 73.3 | 37.9 |
| Full-precision BR | 17.22 | 2.50 | **118.3** | **70.8** | **37.6** |

(d) Ablation study of binarizing different parts

| Binarized Network | Params (M) | OPs (G) | All | Hand | Face |
|---|---|---|---|---|---|
| Body Encoder | 47.78 | 7.45 | 119.8 | 65.9 | 36.7 |
| Hand Encoder | 47.78 | 7.45 | 86.0 | 49.0 | 27.9 |
| Face Encoder | 57.08 | 9.94 | 86.8 | 55.3 | 25.9 |

When compared to the 32-bit full-precision methods, the proposed BiDRN also achieves comparable performance with extremely lower memory and computational cost. For the EHF dataset, BiDRN impressively narrows the *All MPVPEs* gap between full-precision Hand4Whole and binarization methods from **85.9** to just **32.0**. For the AGORA dataset, surprisingly, our BiDRN even surpasses full-precision frameworks ExPose and FrankMocap. Given that AGORA is a more complex and natural dataset (Moon et al., 2022a; Patel et al., 2021), it can better demonstrate the effectiveness of our BiDRN. This also suggests that it will be more valuable to binarize a powerful model (*e.g.,* Hand4Whole), as it may perform better even after a lightweight adaptation.

## 4.3 QUALITATIVE RESULTS

We follow previous work (Moon et al., 2022a; Lin et al., 2023) to show the qualitative results on MSCOCO dataset, as depicted in Figure 7. It can be observed that the 3D human meshes recovered by previous BNN methods cannot even match the 2D images, resulting in completely incorrect results. Conversely, our BiDRN demonstrates a remarkable ability to align with all 2D images, effectively handling even the scenarios set against complex backgrounds, as particularly showcased in the fourth and final rows. Moreover, previous BNN approaches tend to generate wrong rotations, e.g., the third and fifth rows. While our BiDRN keeps the original rotations, as well as capturing more accurate facial expressions and hand poses. Finally, BiDRN exemplifies greater stability compared to traditional BNNs, achieving accurate and consistent estimations across all images. More visual comparisons of EHF and AGORA datasets are shown in supplementary file.

## 4.4 ABLATION STUDY

**Break-down Ablation.** We first establish a baseline with base Local Convolution Residual (LCR). Next, we incrementally introduce Down Scale Residual (DScR), Fusion Up Residual (FUR), Fusion Down Residual (FDR), and Down Sample Residual (DSaR) to improve performance. Notably, our baseline LCR (BaseLCR) achieves an *All MPVPEs* of 139.3, already outperforming the basic BNN (172.2) by a significant margin. As shown in Table 2a, when we successively use DScR, FUR, FDR, and DSaR, the *All MPVPEs* is reduced by 11.5, 1.8, 1.3, and 6.4 respectively. They together reduce the *All MPVPEs* by 21.0 in total with just a few additional Params and OPs, demonstrating the effectiveness of LCR and its four derived modules.

**Pre-activation.** We compare the Hardtanh pre-activation used in our BiDRN with the previous default pre-activation functions ReLU and PReLU. As shown in Table 2b, when replacing ReLU or PReLU with Hardtanh, the *All MPVPEs* can be reduced by 8.5 and 7.6 respectively without additional parameters. This suggests the superiority of the Hardtanh pre-activation in our BiDRN.

**Block Residual.** To study the effect of Block Residual, we remove it from BiDRN, and also compare it with its binarization counterpart. It can be observed from Table 2c that without Block Residual, our method can still achieve 139.6 *All MPVPEs*, which surpasses basic BNN (172.2) with just half of the Params and OPs. When adding the binarized BR, the *All MPVPEs* can be reduced by 19.6, which is a significant improvement with just a few additional Params and OPs. By replacing the binarized BR with full-precision BR, the *All MPVPEs* can be further reduced by 1.7.

Although the improvement of full-precision is not particularly large in quantitative results, the qualitative results in Figure 8 show that full-precision is actually very important for accurate 3D human mesh recovery. It can be observed that only Full-precision BR recovers the accurate hand position and rotation, while Binarized BR only recovers the body well but worse aligns the hands.

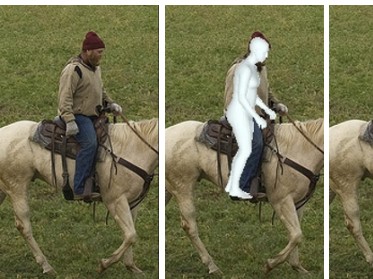 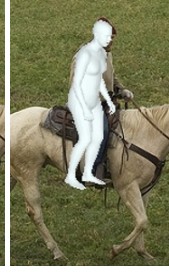 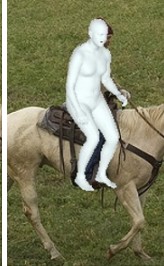

| Image | w/o BR | Binarized BR | Full-precision BR |

Figure 8: Visual comparison of Block Residual ablation study.

**Binarizing Different Networks.** Since body, hand, and face use separate encoder networks, we binarize one of them while keeping the other two as full-precision to study the binarization benefit of different parts. The experimental results are listed in Table 2d, we can observe that **(1)** Binarizing the encoder leads to a corresponding performance drop. However, the *MPVPE* of face is improved when binarizing the Face Encoder, suggesting that the full-precision face encoder has many parameter and operation redundancies, and our binarization method can retain full-precision information well. **(2)** The binarization of Body Encoder also leads to a performance drop of hand and face. In contrast, binarization of the Hand or Face Encoder has little impact on other parts. This suggests that the body encoder is the key point of 3D human mesh recovery since the face and hands boxes are predicted by the body feature. Therefore, the full-precision information on the body feature is more important.

**BoxNet.** To further verify the effectiveness of Binarized BoxNet, we compare it with the full-precision BoxNet. As shown in Table 3, Binarized BoxNet achieves even better performance with much fewer parameters and operations, suggesting that the full-precision BoxNet is redundant and will lead to a performance drop.

Table 3: Ablation study of BoxNet on EHF, where both binarized and full-precision BoxNets are trained with Binarized Block Residual.

| Method | Params (M) | OPs (G) | All | Hand | Face |
|---|---|---|---|---|---|
| Full-precision BoxNet | 21.81 | 1.87 | 130.7 | 78.4 | 40.5 |
| Binarized BoxNet | 11.68 | 1.28 | 125.5 | 75.1 | 39.0 |

### 4.5 COMPARISON TO OTHER COMPRESSION METHODS

**Comparison to Lightweight Model.** An alternative approach to achieve efficiency is to use a smaller but full-precision network, which can also effectively reduce the memory and computational cost. Thus, to demonstrate that our BiDRN can achieve a better efficiency-accuracy trade-off, we

Table 4: Comparison to other compression methods.

| Method | Params (M) | OPs (G) | All | Hand | Face |
|---|---|---|---|---|---|
| Hand4Whole (ResNet-18) | 49.46 | 8.55 | 97.3 | 50.9 | 35.9 |
| BiDRN[h] (Ours) | 47.78 | 7.45 | 86.0 | 49.0 | 27.9 |
| Hand4Whole (L1 pruning) | 25.51 | 2.76 | 146.3 | 79.3 | 45.0 |
| BiDRN (Ours) | 17.22 | 2.50 | 118.3 | 70.8 | 37.6 |

compare it with a lightweight version of the full-precision model Hand4Whole, which replaces the ResNet-50 backbone with ResNet-18. For a fair comparison, we choose BiDRN[h] (second row of Table 2d) that has similar Parameters and Operations. As shown in Table 4, our BiDRN[h] has a significant improvement of *MPVPEs* (13.1%, 3.9%, and 28.7% on All, Hand, and Face respectively) compared to the smaller full-precision model, with even less memory and computational costs.

**Comparison to Pruning Method.** We also compare BiDRN to Hand4Whole with the unstructured pruning method, with L1 norm as criteria. We prune 90% weights of the convolutional layers in body, hands, and face encoders. As shown in Table 4. BiDRN largely outperforms the weight pruning method with fewer parameters and operations.

## 5 CONCLUSION

In this work, we propose BiDRN, a novel BNN-based method for 3D whole-body human mesh recovery. To the best of our knowledge, this is the first work to study the binarization of 3D whole-body human mesh recovery problem. The key to preserving estimation accuracy is to maintain the full-precision information as much as possible. To this end, we present a new binarized unit BiDRB with Local Convolution Residual and Block Residual. Comprehensive quantitative and qualitative experiments demonstrate that our BiDRN significantly outperforms SOTA BNNs and even achieves comparable performance with full-precision 3D whole-body human mesh recovery methods.

ETHICS STATEMENT

The research conducted in the paper conforms, in every respect, with the ICLR Code of Ethics.

REPRODUCIBILITY STATEMENT

We have provided implementation details in Sec. 4. We will also release all the code and models.

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
