# BiDRN: Binarized 3D Whole-body Human Mesh Recovery

## A  Mathematics of Parametric Model SMPL-X

SMPL-X (Pavlakos et al., 2019) utilizes standard vertex-based linear blend skinning with learned corrective blend shape, which is an extension of SMPL (Loper et al., 2023) with fully articulated hands and an expressive face. It contains $N{=}10,475$ vertices and $K{=}54$ joints, including joints for the neck, jaw, eyeballs, and fingers. SMPL-X is a parametric model with pose parameters $\theta$ (consists of $\theta_{body}$, $\theta_{hand}$, and $\theta_{face}$), shape parameters $\beta$, and facial expression parameters $\psi$, defined by

$$M(\theta, \beta, \psi) : \mathbb{R}^{|\theta| \times |\beta| \times |\psi|} \mapsto \mathbb{R}^{3N}. \tag{1}$$

SMPL-X first defines the shape blend, pose blend, and expression blend shape functions as

$$B_S(\beta; \mathcal{S}) = \sum_{n=1}^{|\beta|} \beta_n \mathcal{S}_n, \tag{2}$$

$$B_P(\theta; \mathcal{P}) = \sum_{n=1}^{9K} (R_n(\theta) - R_n(\theta^*))\mathcal{P}_n, \tag{3}$$

$$B_E(\psi; \mathcal{E}) = \sum_{n=1}^{|\psi|} \psi_n \mathcal{E}, \tag{4}$$

where $\mathcal{S}_n, \mathcal{P}_n \in \mathbb{R}^{3N}$ are orthonormal principle components of vertex displacements, $R : \mathbb{R}^{|\theta|} \mapsto \mathbb{R}^{9K}$ is a function mapping the pose vector $\theta$ to a vector of concatenated part-relative rotation matrices, $\theta^*$ is the pose vector of the rest pose. These functions add corrective vertex displacements to the template mesh $\bar{T}$ (Loper et al., 2023) by

$$T_P(\beta, \theta, \psi) = \bar{T} + B_S(\beta; \mathcal{S}) + B_P(\theta; \mathcal{P}) + B_E(\psi; \mathcal{E}). \tag{5}$$

Since different body shapes have a variety of 3D joint locations, the 3D joint locations $J$ is defined as a function of body shape, with the following form:

$$J(\beta) = \mathcal{J}(\bar{T} + B_S(\beta; \mathcal{S})), \tag{6}$$

where $\mathcal{J}$ is a sparse linear regressor that can regress 3D joint locations from mesh vertices.

Finally, the 3D whole-body human mesh is constructed with the above elements as

$$M(\beta, \theta, \psi) = W(T_P(\beta, \theta, \psi), J(\beta), \theta, \mathcal{W}), \tag{7}$$

where $W(\cdot)$ is a standard linear blend skinning function that rotates the vertices in $T_P(\cdot)$ around the estimated joints $J(\beta)$ smoothed by blend weights $\mathcal{W} \in \mathbb{R}^{N \times K}$.

## B  More Details of Our Proposed BiDRN

**Binarized whole-body mesh recovery pipeline.** Given a single RGB human image $I \in \mathbb{R}^{3 \times H \times W}$, our BiDRN extracts body, hands, and face features $F_b$, $F_h$, and $F_f$, where the hands and face input images are cropped and resized from the high-resolution human image $I$ by applying RoIAlign (He et al., 2017). BiDRN replaces the BasicBlock and Bottleneck in ResNet with our proposed BiDRB, which can preserve full-precision information as much as possible in the binarization network. Then following Moon et al. (2022), we utilize these features to estimate the corresponding body, hands, and face parameters $\Theta = \{\theta_{body}, \beta, t, \theta_{hand}, \theta_{face}, \psi\}$ by Pose2Pose architecture and GAP-based regressor. Specifically, $\theta_{body} \in \mathbb{R}^{22 \times 3}$ is the 3D body joint rotation parameters, $\beta \in \mathbb{R}^{10}$ is the body shape parameters, $t \in \mathbb{R}^3$ is the 3D global translation parameters, $\theta_{hand} \in \mathbb{R}^{30 \times 3}$ is the 3D left and right hands joint rotation parameters (each hand contains 15 joints), $\theta_{face} \in \mathbb{R}^3$ is the 3D

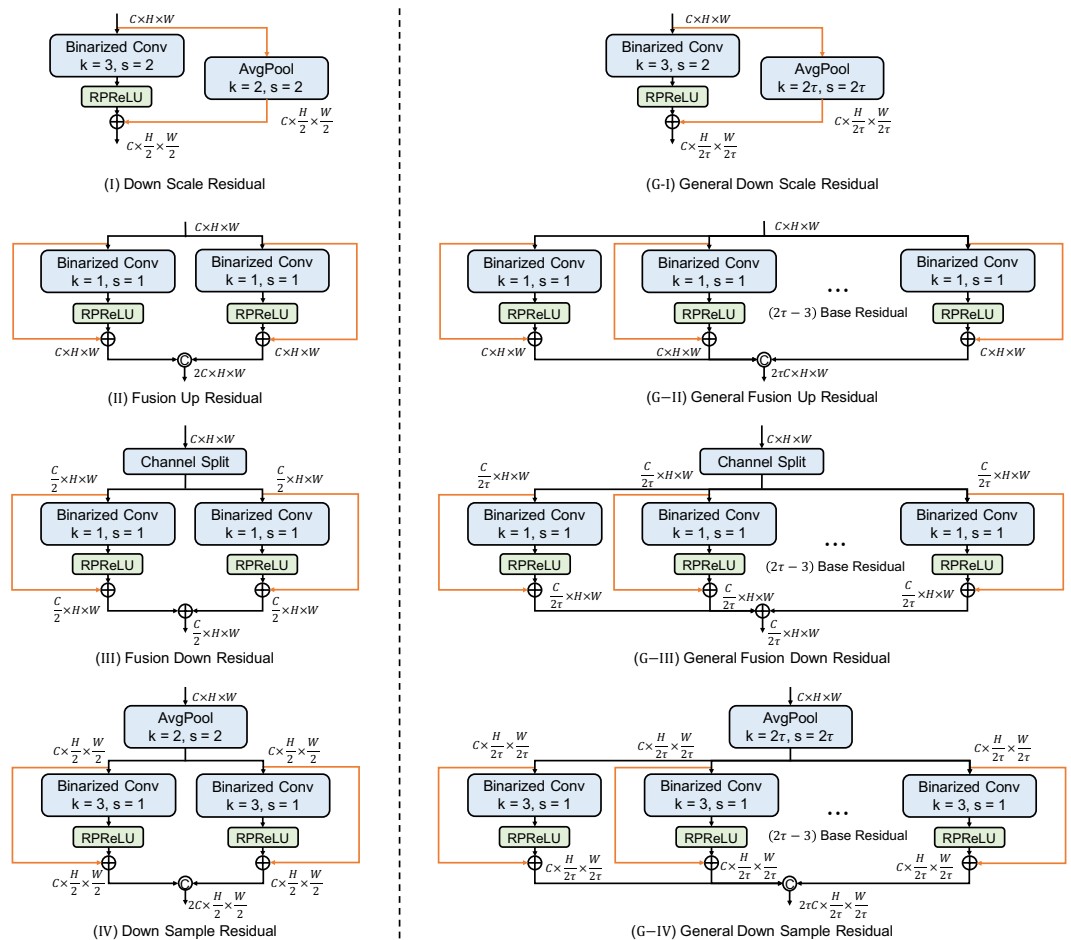

Figure 1: Generalization of four redesign convolutional modules, including **(G-I)** General Down Scale Residual (G-DScR), **(G-II)** General Fusion Up Residual (G-FUR), **(G-III)** General Fusion Down Residual (G-FDR), and **(G-IV)** General Down Sample Residual (G-DSaR). The orange arrow denotes the full-precision information flow. For simplicity, both batch normalization and Hardtanh pre-activation are omitted.

jaw rotation parameters and $\psi \in \mathbb{R}^{10}$ is the facial expression parameters. Finally, these parameters $\Theta$ are fed into a SMPL-X layer (Pavlakos et al., 2019) to obtain the 3D whole-body human mesh.

**General Local Convolution Residual.** We redesign four kinds of convolution modules (DScR, FUR, FDR, and DSaR) based on the Local Convolution Residual (LCR) to tackle the dimension mismatched problem. Note that we have already described the basic condition (double or half the dimension) in the main paper. Here we further consider the more complex and general conditions, as shown in Figure 1. Specifically, we introduce a reshape-parameter $\tau \in \mathbb{Z}^{+}$ and generalize these four convolution modules to the conditions where channel-wise and spatial-wise dimensions are changed based on $2\tau$. We depict and formulate the general versions of them as below.

General Down Scale Residual (G-DScR) compresses the spatial dimension by $1/2\tau$. To match the output dimension, the full-precision activation is first fed into an average pooling function with *kernel-size* $2\tau$ and *stride* $2\tau$. Then it is added to the output of binarized Down Scale convolution as

$$\boldsymbol{o}' = \mathrm{BatchNorm}(\mathrm{RPReLU}(\boldsymbol{o}) + \mathrm{AvgPool}_{\tau}(\boldsymbol{a}_f)), \qquad (8)$$

where $\boldsymbol{o}', \boldsymbol{o} \in \mathbb{R}^{C \times \frac{H}{2\tau} \times \frac{W}{2\tau}}, \boldsymbol{a}_f \in \mathbb{R}^{C \times H \times W}$.

General Fusion Up Residual (G-FUR) increases the channel dimension by $2\tau$ times. Therefore, we first feed the activation to $2\tau$ distinct Base Residuals, and then concatenate the output of them as

$$\boldsymbol{o}' = \text{BatchNorm}(\text{Concat}(\boldsymbol{o}'_1, \boldsymbol{o}'_2, \ldots, \boldsymbol{o}'_{2\tau})), \tag{9}$$

where $\boldsymbol{o}' \in \mathbb{R}^{2\tau C \times H \times W}, \boldsymbol{o}'_1, \boldsymbol{o}'_2, \ldots, \boldsymbol{o}'_{2\tau} \in \mathbb{R}^{C \times H \times W}$.

General Fusion Down Residual (G-FDR) compresses the channel dimension by $1/2\tau$. As it is the inverse of G-FUR, we first channel-wise split and feed the activation to $2\tau$ Base Residuals, and then sum up their output as

$$\boldsymbol{o}' = \text{BatchNorm}(\sum_{k=1}^{2\tau} \boldsymbol{o}'_k), \tag{10}$$

where $\boldsymbol{o}' \in \mathbb{R}^{\frac{C}{2\tau} \times H \times W}, \boldsymbol{o}'_1, \boldsymbol{o}'_2, \ldots, \boldsymbol{o}'_{2\tau} \in \mathbb{R}^{\frac{C}{2\tau} \times H \times W}$.

General Down Sample Residual (G-DSaR) compresses the spatial dimension by $1/2\tau$ while increasing the channel dimension by $2\tau$ times. As it is the mix of G-DScR and G-FUR, we first apply the average pooling with reshape-parameter $\tau$ to the activation and then concatenate the output of $2\tau$ distinct Base Residuals as

$$\boldsymbol{o}'_k = \text{BaseLCR}_k(\text{AvgPool}_\tau(\boldsymbol{a}_f)), \tag{11}$$

$$\boldsymbol{o}' = \text{BatchNorm}(\text{Concat}(\boldsymbol{o}'_1, \boldsymbol{o}'_2, \ldots, \boldsymbol{o}'_{2\tau})), \tag{12}$$

where $\boldsymbol{o}' \in \mathbb{R}^{2\tau C \times \frac{H}{2\tau} \times \frac{W}{2\tau}}, \boldsymbol{o}'_1, \boldsymbol{o}'_2, \ldots, \boldsymbol{o}'_{2\tau} \in \mathbb{R}^{C \times \frac{H}{2\tau} \times \frac{W}{2\tau}}, \boldsymbol{a}_f \in \mathbb{R}^{C \times H \times W}, k = 1, 2, \ldots, 2\tau$, and BaseLCR$_k$ denotes the $k$-th Base Local Convolution Residual.

Note that $\tau$=1 is the basic condition in our paper and the proposed BiDRN involves the conditions $\tau$=1 and $\tau$=2. Compared to previous work, we further extend and address the dimension mismatched problem in general conditions.

**Mapping Table.** As shown in Table 1, we present a mapping table from ResNet-18 and ResNet-50 backbones to our proposed modules in BiDRN.

Table 1: Mapping table from ResNet-18 and ResNet-50 backbones to our BiDRN.

| ResNet-18 (face) | | | ResNet-50 (body and hands) | | |
|---|---|---|---|---|---|
| **from** | **to** | | **from** | **to** | |
| conv1 | conv1 | | conv1 | conv1 | |
| bn | bn | | bn | bn | |
| relu | tanh | | relu | tanh | |
| MaxPool | MaxPool | | MaxPool | MaxPool | |
| AvgPool | AvgPool | | AvgPool | AvgPool | |
| BasicBlock_1 | LCR
LCR | BR | Bottleneck_1 | LCR + FUR
FDR + FUR
FDR + FUR | BR |
| BasicBlock_2 | DSaR + LCR
LCR | BR | Bottleneck_2 | FDR + DScR + FUR
FDR + FUR
FDR + FUR
FDR + FUR | BR |
| BasicBlock_3 | DSaR + LCR
LCR | BR | Bottleneck_3 | FDR + DScR + FUR
FDR + FUR
FDR + FUR
FDR + FUR
FDR + FUR
FDR + FUR | BR |
| BasicBlock_4 | DSaR + LCR
LCR | BR | Bottleneck_4 | FDR + DScR + FUR
FDR + FUR
FDR + FUR | BR |

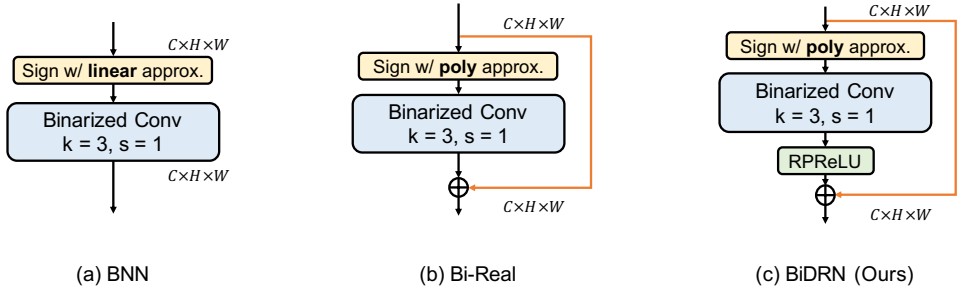

Figure 2: A comparison of **Identity Layer** (where input and output share the same dimensions) between BNN, Bi-Real, and our proposed BiDRN, with batch normalization omitted for clarity.

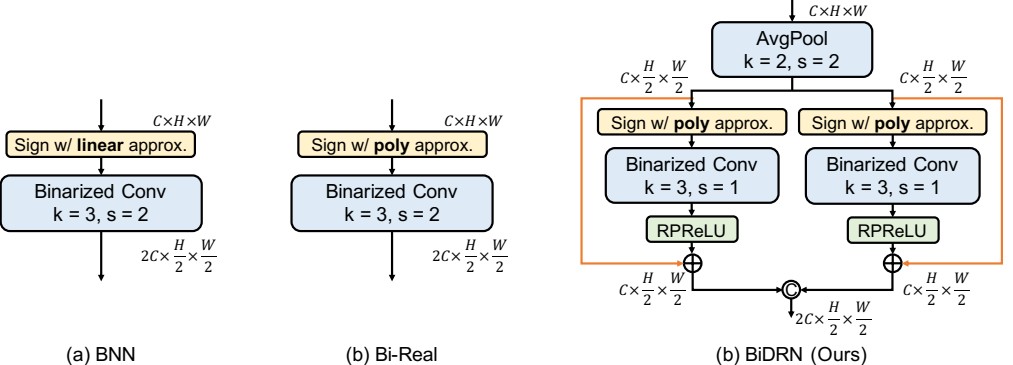

Figure 3: A comparison of **Down Sample Layer** (where input and output have different dimensions) among BNN, Bi-Real, and our proposed BiDRN, with batch normalization omitted for clarity. Since the output dimension differs from the input, the residual connection in Bi-Real cannot be applied under these conditions. However, our BiDRN introduces a Down Sample Residual (DSaR) mechanism, which effectively bypasses full-precision activation.

**Architecture Comparison.** We provide an architectural comparison of BNN, Bi-Real, and our proposed BiDRN to highlight their differences. Specifically, **Identity Layer** comparisons are shown in Figure 2, where BiDRN incorporates RPReLU to redistribute the binarized output before applying the residual connection. Additionally, in Figure 3, we compare the **Down Sample Layer**, where the input and output dimensions differ. In Bi-Real, the bypass mechanism used in the **Identity Layer** cannot be applied to the **Down Sample Layer** due to the dimension mismatch. In contrast, BiDRN redesigns the Down Sample module to integrate the residual connection and retain as much full-precision information as possible.

## C  EFFICIENCY COMPARISON WITH SOTA FULL-PRECISION METHODS

We compare the parameters (Params) and operations (OPs) of several full-precision methods, including ExPose, PIXIE, and Hand4Whole, with our proposed BiDRN in Table 2. It can be observed that the Params of BiDRN is just **12.7%**, **8.9%**, **22.1%** of ExPose, PIXIE, and Hand4Whole respectively. Similarly, when considering OPs, BiDRN requires merely **8.8%**, **7.3%**, **14.8%** of them respectively. Note that BiDRN even outperforms ExPose on the AGORA dataset (see Table 1 of the main paper).

Table 2: Params and OPs comparison between full-precision methods and our BiDRN.

| Method | Params (M) | OPs (G) |
|---|---|---|
| ExPose (Choutas et al., 2020) | 135.8 | 28.5 |
| PIXIE (Feng et al., 2021) | 192.9 | 34.3 |
| Hand4Whole (Moon et al., 2022) | 77.8 | 16.9 |
| BiDRN (Ours) | 17.2 | 2.5 |

## D   MORE EXPERIMENTAL RESULTS

**Smooth Function for Sign.**   We conduct an experiment to compare the 2nd-order approximation with the tanh approximation for Sign in Table 3. The results show that the 2nd-order approximation performs better, leading us to adopt the 2nd-order smooth function.

**Redistribution function.**   We conduct an experiment to compare PRReLU and LeakyReLU redistribution functions in Table 4. We find that the performance degrades if using LeakyReLU. Considering that the Params and OPs are almost the same, it will be better to use RPReLU as redistribution function.

Table 3: Comparison between 2nd-order approximation and tanh approximation.

| Method | Params (M) | OPs (G) | All | Hand | Face |
|---|---|---|---|---|---|
| BiDRN (Ours w/ 2nd-order approx.) | 17.22 | 2.50 | **118.3** | **70.8** | **37.6** |
| BiDRN (Ours w/ tanh approx.) | 17.22 | 2.50 | 122.7 | 72.1 | 39.1 |

Table 4: Comparison between PRReLU and LeakyReLU redistribution functions.

| Method | Params (M) | OPs (G) | All | Hand | Face |
|---|---|---|---|---|---|
| BiDRN (Ours w/ PRReLU) | 17.22 | 2.50 | **118.3** | **70.8** | **37.6** |
| BiDRN (Ours w/ LeakyReLU) | 17.21 | 2.50 | 130.6 | 75.2 | 41.1 |

**Generalizability of our BiDRN.**   To demonstrate the generalizability of our proposed BiDRN, we apply it to the classification task on the MNIST dataset, as shown in Table 5. We first performed experiments with a full-precision ResNet-18 model, which is commonly used for MNIST classification. We then binarized ResNet-18 using BNN's binary convolution layers, as well as BiDRN's LCR, BR, and four derived modules. It can be observed that BiDRN can be extended to other tasks with significant improvement as well. Besides the classification task, we also apply BiDRN to a keypoint prediction task using the COCO dataset. Additionally, to validate the generalizability of our proposed layer to other models, we integrated our modules into MobileNetv1. As shown in Table 6, the results indicate that BiDRN is also effective for the 2D keypoint prediction task. We believe these additional results demonstrate that our BiDRN method excels in simpler 2D/3D pose estimation tasks.

Table 5: BiDRN on classification task.

| Method | Params (M) | OPs (M) | Test error rate ↓ |
|---|---|---|---|
| ResNet-18 (full-precision) | 11.2 | 1826.0 | 0.72% |
| BNN | 0.4 | 40.7 | 2.63% |
| BiDRN (Ours) | 0.4 | 40.7 | 1.15% |

**Comparison to BBCU.**   We conduct an experiment to compare BiDRN with the recent binarization method BBCU (Xia et al., 2023), which is also applied in ResNet-like models. As shown in Table 7, our BiDRN also largely outperforms BBCU with fewer parameters and operations.

## E   MORE VISUALIZATION

We compare the mesh quality of our BiDRN with 7 SOTA BNN-based methods (including BNN (Hubara et al., 2016), XNOR (Rastegari et al., 2016), DoReFa (Zhou et al., 2016), Bi-Real (Liu et al., 2018), ReActNet (Liu et al., 2020), ReCU (Xu et al., 2021b), and FDA (Xu et al., 2021a)) on the EHF dataset ( Figure 4) and AGORA ( Figure 5) dataset. Due to the page limit, the qualitative result of break-down ablation is shown in Figure 6.

**Qualitative comparisons on EHF.**   Images in the EHF dataset contain a single subject performing a variety of interesting body poses, hand gestures, and facial expressions. Yet, the background of the images is dark, which contributes to a lower contrast between the subject and the surroundings. Therefore, the overall performance of the binarization methods is not good. It can be observed that previous SOTA BNNs can hardly match the hands' positions, BNN and FDA even generate totally

Table 6: BiDRN on 2D keypoint prediction task.

| Method | Params (M) | OPs (M) | MPJPE↓ |
|---|---|---|---|
| MobileNetv1 (full-precision) | 3.2 | 583.3 | 176.5 |
| BNN | 0.2 | 17.9 | 338.3 |
| BiDRN (Ours) | 0.2 | 17.9 | 188.3 |

Table 7: Comparison between BBCU and BiDRN.

| Method | Params (M) | OPs (G) | All | Hand | Face |
|---|---|---|---|---|---|
| BBCU | 22.04 | 2.68 | 164.9 | 89.7 | 50.3 |
| BiDRN (Ours) | 17.22 | 2.50 | **118.3** | **70.8** | **37.6** |

wrong rotations (the first and final row). In contrast, our BiDRN aligns the body, hands, and face better and achieves a more stable performance among all examples.

**Qualitative comparisons on AGORA.** AGORA is a synthetic dataset with limited actions, such as walking, sitting, and taking phones. It can be observed that the proposed BiDRN outperforms all SOTA BNN-based algorithms in terms of whole-body poses, hand poses, and global orientations. In contrast, previous BNN methods can hardly align the body in simple situations (the second row).

**Qualitative comparisons of break-down ablation.** As shown in Figure 6, the BaseLCR can already match the 2D image roughly, and successively adding these four modules can fine-tune the body rotation, hand position, and leg angle step by step.

Table 8: Real inference time on hardware.

| Method | Params (M) | OPs (G) | Frame | Infer Time (ms) |
|---|---|---|---|---|
| Hand4Whole | 77.84 | 16.85 | TF Lite | 54.8 |
| BiDRN (Ours) | 17.22 | 2.50 | daBNN (Zhang et al., 2019) | 12.1 |

## F    REAL INFERENCE TIME

The comparison of inference time on real hardware may be a more intuitive way to show efficiency. However, the general-purpose inference libraries for GPUs (e.g., TensorRT) do not support binarized convolutions and linear projections, which require specific hardware. We lack such hardware, and this is the problem we encounter. Fortunately, people who study quantification provide a benchmark (Zhang et al., 2019) for estimating inference time to alleviate this problem. According to Zhang et al. (2019), we calculate the inference time (on ARM devices) of our binarization in Table 8, where the inference time of our BiDRN is only **22.1%** of the full-precision Hand4Whole. Complementary to the theoretical analysis, this suggests the efficiency of our BiDRN from the practical perspective.

## G    ACADEMIC AND SOCIAL IMPACT

3D whole-body human mesh recovery is one of the core technologies to understand human behaviors, which is widely applied in AR/VR (Wang et al., 2021), sign language recognition (Camgoz et al., 2020) and emotion recognition (Lin et al., 2023). With the rapid growth in the number of pictures and videos, how to efficiently recover 3D whole-body human mesh is worth studying. Our BiDRN is able to achieve more efficient and accurate 3D mesh estimations than existing SOTA BNN-based algorithms. We hope this work will promote further research on efficient 3D whole-body human mesh recovery.

Currently, both 3D whole-body human mesh recovery techniques and our proposed BiDRN do not present any negative foreseeable societal consequences.

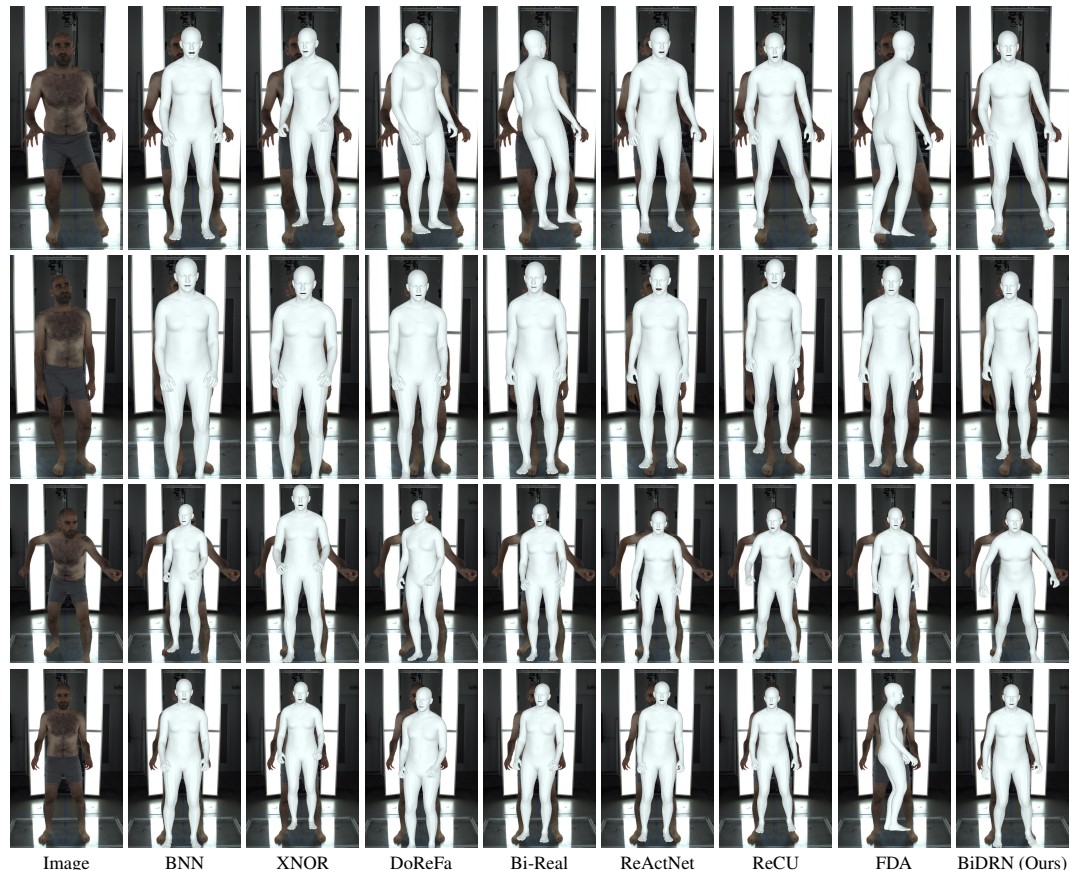

| Image | BNN | XNOR | DoReFa | Bi-Real | ReActNet | ReCU | FDA | BiDRN (Ours) |

Figure 4: Qualitative comparison between seven existing SOTA BNN-based approaches and our newly proposed BiDRN on the EHF (Pavlakos et al., 2019) dataset.

## H LIMITATION AND FUTURE WORK

Currently, our BiDRN does not improve the estimation accuracy of different parts uniformly. Specifically, the enhancement of hand estimation is smaller than that of the body and face. It is worth studying how to further improve hand estimation accuracy in future work. Meanwhile, with the emergence of powerful Transformer-based models (e.g.OSX (Lin et al., 2023)), how to extend our binarization method to these Transformer-based architectures requires further study in future work as well.

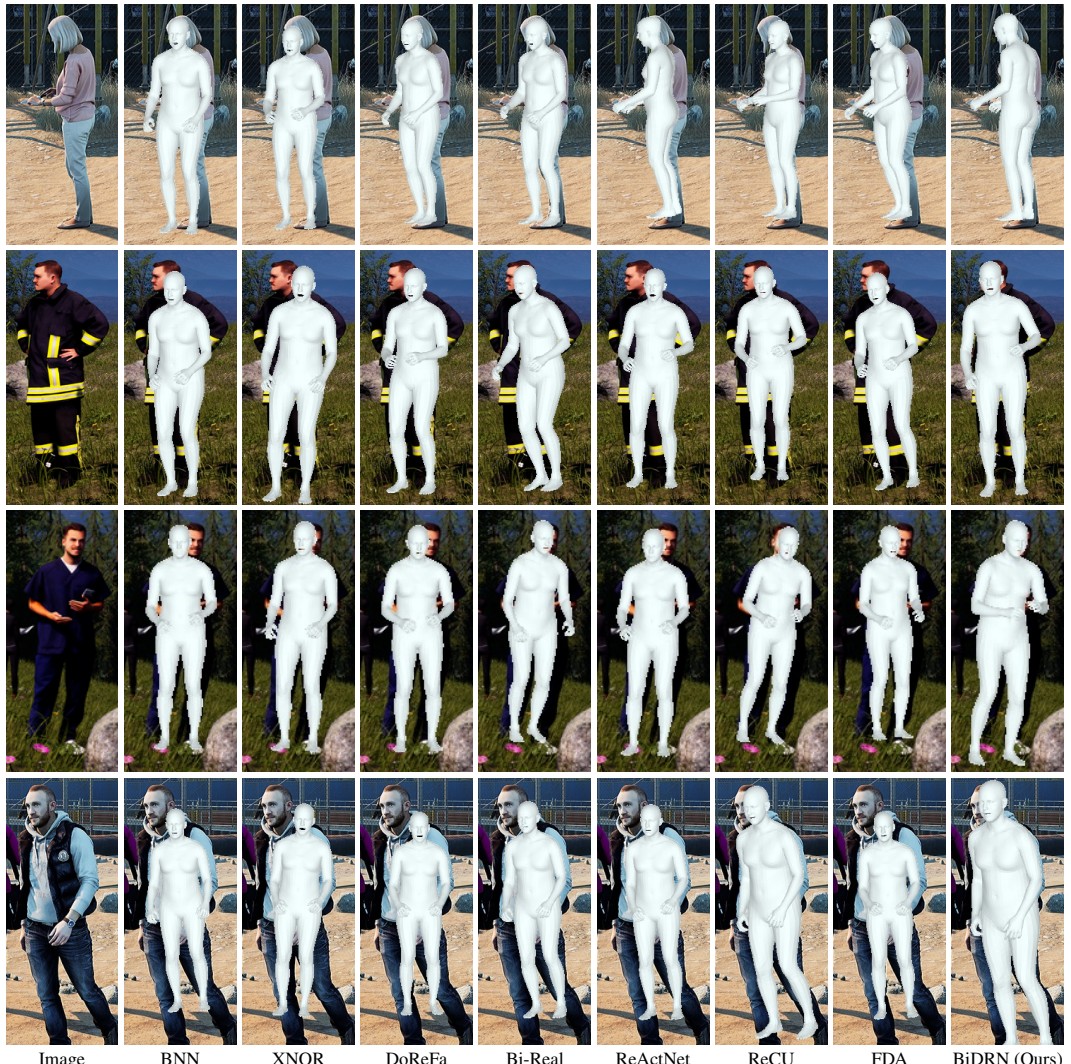

Figure 5: Qualitative comparison between seven existing SOTA BNN-based approaches and our newly proposed BiDRN on the AGORA (Patel et al., 2021) dataset.

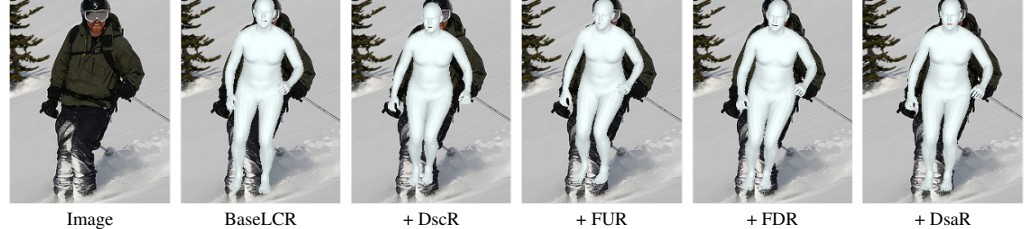

Figure 6: Visual comparison of break-down ablation study on EHF dataset.