# OpenReview forum: "BiDRN: Binarized 3D Whole-body Human Mesh Recovery"
_ICLR.cc/2025/Conference — Submitted to ICLR 2025_

### Official Review · Reviewer_zufd · 2024-10-27

**Soundness:** 3
**Presentation:** 3
**Contribution:** 3
**Rating:** 8
**Confidence:** 4

**Summary:**

In this work, the authors focus on the 3D HMR (human mesh recovery) task. They propose a method named BiDRN based on a Binarized Dual Residual Network. They propose to use a binarized boxnet followed by three BiDRNs for the face, hand, and body, respectively. These features are then projected to SMPL-X parameters to animate the human object. As suggested by the authors, they are the first to propose a binarized neural network for the HMR task. The authors have conducted extensive comparisons on EHF and AGORA benchmarks and compared both binarized and non-binarized methods.

**Strengths:**

+ The proposal of BiDRN is one of the first implementations of a Binarized Neural Network for the HMR task. The motivation is solid (due to rendering time requirements for VR), and the paper is well-written and easy to follow.
+ The authors have conducted comprehensive comparisons with other methods, including both BNN-based and non-BNN-based methods. They show significant improvement over other BNN-based methods, while performance is close to float32 networks, with a huge reduction in computational requirements.
+ The overall improvement is validated with extensive ablation results.

**Weaknesses:**

This work is interesting, and I have no major concerns. For further improvement, here are some suggestions:

+ For the results shown in Table 1, please consider including more recent BNN-based methods. Currently, the latest work in the table is from 2021, which is three years old. Including more recent work would make the comparison more reasonable, as the BNN domain has advanced in recent years (as stated in the related works).
 + As BNNs significantly reduce computational requirements, it would be helpful to check the FPS rate for a video for the BNN network compared to the original network, which could better illustrate whether the BNN-based model can achieve real-time HMR.

**Questions:**

If authors are able to provide the FPS rate for the reconstruction for a video, along with the resolution of the frames and computational resources, it can be very helpful to further validate how fast BNN can achieve for HMR task.

---

> ### Author Response · Authors · 2024-11-22
> **Response to Reviewer zufd (denoted as R4)**
>
> `Q4-1:` For the results shown in Table 1, please consider including more recent BNN-based methods. Currently, the latest work in the table is from 2021, which is three years old. Including more recent work would make the comparison more reasonable, as the BNN domain has advanced in recent years (as stated in the related works).
>
> `A4-1:` Thank you for your valuable suggestion. Unfortunately, to the best of our knowledge, more recent BNN-based methods focus on specific domains and are not designed as general-purpose solutions. For instance, BBCU [ref1] is tailored for image restoration, while BiLLM [ref2] targets large language models. Nevertheless, your suggestion is valuable, and we attempted to **adapt BBCU to the HMR task** by applying its Basic Binary Conv Unit to the backbone. The results demonstrate that our BiDRN significantly outperforms it.
>
> | **Method**                          | **Params (M)** | **OPs (G)** | **MPVPE (All)$\downarrow$**  | **MPVPE (Hand)$\downarrow$**  | **MPVPE (Face)$\downarrow$**  |
> | ------------------------------- | ---------- | ------- | ------- | ------- | ------- |
> | BBCU                            | 22.04      | 2.68    | 164.9   | 89.7 | 50.3 |
> | BiDRN (Ours)                    | 17.22        | 2.50    | 118.3   | 70.8 | 37.6 |
>
> [ref1] Basic Binary Convolution Unit For Binarized Image Restoration Network, ICLR, 2023.
>
> [ref2] BiLLM: Pushing the Limit of Post-Training Quantization for LLMs, ICML, 2024.
>
> `Q4-2:` As BNNs significantly reduce computational requirements, it would be helpful to check the FPS rate for a video for the BNN network compared to the original network, which could better illustrate whether the BNN-based model can achieve real-time HMR.
>
> `A4-2:` Thank you for your suggestion. We have provided the inference time comparison in **Table 7 of the Appendix**, where our BiDRN achieves impressive acceleration compared to Hand4Whole. The deployment to support video inference requires more engineering work, and we will incorporate this into our future work.
>
> `Q4-3:` If authors are able to provide the FPS rate for the reconstruction for a video, along with the resolution of the frames and computational resources, it can be very helpful to further validate how fast BNN can achieve for HMR task.
>
> `A4-3:` Thank you for your suggestion. Adapting the method for video reconstruction involves significant engineering effort. Due to time and hardware constraints, we plan to address this as part of our future work.

---

> > ### Comment · Reviewer_zufd · 2024-11-25
> >
> > Thank you for your response, I will hold my original rating.

---

> > > ### Author Response · Authors · 2024-11-28
> > >
> > > Thank you for your response and for maintaining a positive rating.

---

### Official Review · Reviewer_Ywt4 · 2024-11-03

**Soundness:** 3
**Presentation:** 2
**Contribution:** 1
**Rating:** 3
**Confidence:** 4

**Summary:**

This paper investigates the binarized network for whole-body 3D human mesh recovery. The motivation is to reduce the high computational cost of existing multi-stage pipeline, so that the model can efficiently run on mobile devices. While directly binarizing the backbone causes severe degradation on precision of mesh regression. Built on the baseline, Hand4Whole, this paper redesigns the architecture of backbone, bounding box detector, and decoding heads to reduce the computational costs while alleviating the reduction in precision.  The experiments are conducted on EHF and AGORA to testify the effectiveness of the proposed designs, compared with other common designs. Compared to the previous binarization design, ReCU, the proposed method reduces 17.1% and 4.5% on EHF and AGORA in terms of whole-body MPJPE, meanwhile, reducing the computational costs to 27.8% of the original Params and 15.6%  of the original OPs.

**Strengths:**

1.	This paper addresses the underexplored challenge of enhancing computational efficiency in whole-body 3D human mesh recovery.
2.	Unlike previous binarization methods, the proposed approach mitigates degradation in mesh regression to an obvious extent.
3.	The reduction in model parameters and operations is substantial, making this method valuable for deployment on mobile devices.

**Weaknesses:**

This paper introduces a binarization model based on the state-of-the-art multi-stage method, Hand4Whole. The experiments and binarization techniques are tailored specifically to optimize this baseline. However, it remains unclear whether the proposed binarization approach is compatible with more efficient one-stage methods, such as MultiHMR or AiOS, which may offer greater potential for faster performance on mobile devices. If the proposed binarization method cannot be applied to other whole-body HMR methods, its novelty and contribution to the field would be limited.

The challenges are described as engineer problems, such as the quality of backbone features or dimension mismatching. The motivation behind all the binarization designs hasn’t been discussed in an obvious way.

**Questions:**

The scientific value of the proposed method might be determined by whether the proposed method can be generalized to the other HMR methods, especially the one-stage methods, AiOS / MultiHMR, which adopts a very different architecture.

---

> ### Author Response · Authors · 2024-11-22
> **Response to Reviewer Ywt4 (denoted as R3)**
>
> `Q3-1:` This paper introduces a binarization model based on the state-of-the-art multi-stage method, Hand4Whole. The experiments and binarization techniques are tailored specifically to optimize this baseline. However, it remains unclear whether the proposed binarization approach is compatible with more efficient one-stage methods, such as MultiHMR or AiOS, which may offer greater potential for faster performance on mobile devices. If the proposed binarization method cannot be applied to other whole-body HMR methods, its novelty and contribution to the field would be limited.
>
> `A3-1:` Thank you for your suggestion. We add additional explanations and experiments below.
> 1. Our **BiDRN is compatible with one-stage methods** as well. Since the majority of parameters in these methods reside in the backbone, the LCR and BR techniques, along with the four redesigned modules introduced by BiDRN, can be adapted for use in their backbones.
> 2. Although our method is compatible with both MultiHMR and AiOS, resource constraints (as AiOS requires substantial training resources, including at least 16 V100 GPUs) prevented us from conducting experiments on the AiOS model. Nevertheless, we followed your suggestion and **applied our BiDRN to MultiHMR**. For a fair comparison, we replaced the backbone of MultiHMR from ViT to the convolution-based ConvMixer, as all the binarization methods discussed in this paper are based on convolution. Due to time constraints, we have not yet completed all experiments. At this stage, we present the results obtained after 70,000 iterations and will include a detailed discussion in our paper once the training is finalized.
>
> | **Method**                          | **\#Iteration** | **EHF** | | **Bedlam** | |
> |---------------------------------|---------|---------|---------|---------|---------|
> | | |**PVE$\downarrow$**  | **PAPVE$\downarrow$**  |**PVE$\downarrow$**  | **PAPVE$\downarrow$**  |
> | BNN                             | 70,000        | 344.9 | 168.2 | 292.4 | 151.1 |
> | BiDRN (Ours)                    | 70,000        | 302.7 | 137.1 | 270.2 | 126.5 |
>
> `Q3-2:` The challenges are described as engineer problems, such as the quality of backbone features or dimension mismatching. The motivation behind all the binarization designs hasn’t been discussed in an obvious way.
>
> `A3-2:` Thank you for your feedback. We provide additional insights into our method to better clarify its motivation:
> 1. The binarization process often results in significant **information loss**, making the full-precision input particularly valuable under such conditions. To address this, we introduced the Local Convolution Residual (LCR) and Block Residual (BR) mechanisms, designed to **bypass and retain full-precision information**. This approach is also inspired by residual connections in full-precision networks, which facilitate training and enhance image feature representation.
> 2. Dimension mismatches are almost unavoidable, as many networks include operations that alter the shape of hidden features. These **mismatches hinder the flow of full-precision information**. To overcome this limitation, we redesigned these modules to ensure compatibility with LCR, enabling seamless propagation of full-precision information from input to output.
>
> `Q3-3:` The scientific value of the proposed method might be determined by whether the proposed method can be generalized to the other HMR methods, especially the one-stage methods, AiOS / MultiHMR, which adopts a very different architecture.
>
> `A3-3:` Thank you for your feedback. The proposed BiDRN **can be extended to one-stage HMR methods**, as these methods also rely on a feature extraction backbone. Consequently, the LCR, BR, and the four redesigned modules can be seamlessly integrated into their backbone architecture. **We have responded to another similar question, `Q3-1`. Please refer to `A3-1` for more details.**

---

> > ### Comment · Reviewer_Ywt4 · 2024-11-26
> > **Thanks, but the application is limited due to the inability to support ViT-based backbones.**
> >
> > Thanks for further extending the experiments on MultiHMR, but the generalization of the proposed method has not been directly and effectively proved.
> >
> > The new experiment replaces the backbone of MultiHMR with a convolution-based ConvMixer and shows obvious improvement compared with the baseline method, BNN, at 70K iterations. If they share the same hyper-parameter configuration. Then this new experiments might reveal that the proposed binarization designs could be compatiable with the other HMR head modules, like MultiHMR. Compared with just optimizing one model, Hand4Whole, this could be helpful.
> >
> > However, it also shows that the proposed binarization designs focus on the model with convolution backbones. While the mainstream models mostly adopt ViT backbones, such as HMR2, TokenHMR, MultiHMR, etc. Especially, experiments in MultiHMR and HMR2 compare the performance of using typical convolution-based and ViT-based backbone. The performance gap is quite obvious. Therefore, can't support ViT-based backbone limits the scope of application. Given all these concerns, I will keep the original rate.

---

> > > ### Author Response · Authors · 2024-11-28
> > >
> > > Thank you for taking the time to review our response. Below, we provide further clarification regarding the intention and scope of our method.
> > >
> > > 1. While many models have adopted ViT backbones, newly developed models like AiOS (as you previously suggested) continue to utilize ResNet backbones. This highlights that convolutional networks remain popular and far from obsolete.
> > > 2. We appreciate your suggestion to support ViT-based backbones. However, our objective is not to develop a general binarized framework that supports all types of HMR backbones, as this would be an overly broad scope for a single work. Supporting CNN-based backbones already covers a wide range of methods, including the latest approach, AiOS. Incorporating ViT backbones would require significant changes and we plan to explore it in the future work.

---

### Official Review · Reviewer_kuYg · 2024-11-04

**Soundness:** 3
**Presentation:** 3
**Contribution:** 2
**Rating:** 5
**Confidence:** 4

**Summary:**

This paper introduces a binarized network for 3D whole-body recovery from images. The core design of the proposed network is a binarized module consisting of (i) the local convolution residual with hardtanh pre-activation to alleviate binarization failures, and (ii) the block residual with a full-precision shortcut to maintain information. The proposed network is compared with several general binarization methods on two public datasets. The experimental results show that the proposed network outperforms the general binarization methods.

**Strengths:**

\+ The presentation of this paper is good and easy to follow.

\+ This paper provides a good engineering solution for efficient human mesh recovery. It raises the performance of previous binary networks on the EHF and AGORA datasets.

**Weaknesses:**

\- The proposed method is not clarified clearly. In the down sample residual block (Figure 4, left), why the dual binarized convolutions with the same inputs, kernel sizes, and strides are needed? How does it differ from a grouped binarized convolution (i.e., with 2 groups)? As for the FullPrecision convolution in the block residual, its kernel size and stride are 1 and 2, respectively. Will this result in checkboard artifacts?

\- None of the competitors is designed for human mesh recovery. I wonder whether the proposed method still has significant advantages in efficiency compared with those tailored for this task (in terms of GPU memory and FPS), such as [1-2].

[1] Dou, Zhiyang, et al. Tore: Token reduction for efficient human mesh recovery with transformer. ICCV 2023.

[2] Zheng, Ce, et al. Potter: Pooling attention transformer for efficient human mesh recovery. CVPR 2023.

\- The qualitative results show that some meshes predicted by the proposed method do not align with images well (e.g., hands and faces in Figure 7).

**Questions:**

I have a few concerns that I wish could be addressed. I may change my decision after reading the rebuttal and other reviewers' comments.

Q1: It would be better to include the results of the baseline (Hand4whole) in the visual examples.

Q2: Why binarized deconvolution and linear layers even yield better performance (L345-346)?

Q3: How is the prediction head of SMPLX parameters binarized?

Q4: A visual architecture comparison with other binarized methods could help to capture the gist and novelty of the proposed design, especially when previous methods have similar components, e.g., Bi-real also adopts a piecewise polynomial function and shortcuts.

Typos: "e.g.the" -> "e.g., the", "Hands regions" -> "Hand regions".

---

> ### Author Response · Authors · 2024-11-22
> **Response to Reviewer kuYg (denoted as R2) part 1**
>
> `Q2-1:` The proposed method is not clarified clearly. In the down sample residual block (Figure 4, left), why the dual binarized convolutions with the same inputs, kernel sizes, and strides are needed? How does it differ from a grouped binarized convolution (i.e., with 2 groups)? As for the FullPrecision convolution in the block residual, its kernel size and stride are 1 and 2, respectively. Will this result in checkboard artifacts?
>
> `A2-1:` Thank you for your question. We add more explanations below.
> 1. As shown in the left part of Figure 4, the outputs of dual binarized convolutions are concatenated because the Down Sample doubles the channel dimension. To integrate it with LCR, we must **ensure that the input and output dimensions of the binarized convolutions remain consistent**. Therefore, we use two binarized convolutions, each incorporating LCR, and concatenate their outputs along the channel dimension.
> 2. A grouped binarized convolution with two groups cannot bypass the full-precision input. However, the Down Sample Residual effectively integrates with LCR, thereby preserving as much full-precision information as possible.
> 3. Checkerboard artifacts typically arise during upsampling operations, such as transposed convolutions. However, the convolution in the block residual performs downsampling. Furthermore, no checkerboard artifacts were observed during our experiments.
>
> `Q2-2:` None of the competitors is designed for human mesh recovery. I wonder whether the proposed method still has significant advantages in efficiency compared with those tailored for this task (in terms of GPU memory and FPS), such as [1-2].
> [1] Dou, Zhiyang, et al. Tore: Token reduction for efficient human mesh recovery with transformer. ICCV 2023.
> [2] Zheng, Ce, et al. Potter: Pooling attention transformer for efficient human mesh recovery. CVPR 2023.
>
> `A2-2:` Thank you for your suggestion. We explain it below.
> 1. Unfortunately, both **[1] and [2] are designed only for the SMPL model**, whereas the baseline Hand4Whole and our BiDRN are tailored for the more advanced SMPL-X model. Since [1] and [2] cannot be adapted to HMR based on the SMPL-X model, we believe it would not be a fair comparison, as the SMPL-X model requires estimating more parameters for HMR tasks.
> 2. To ensure a fair comparison, we applied other efficient methods to Hand4Whole, including the lightweight backbone and pruning techniques outlined in **Table 4 of the main paper**. The results demonstrate that our BiDRN achieves superior performance while utilizing fewer parameters and operations.
>
> | Method                    | Params (M) | OPs (G) | All  | Hand | Face |
> |---------------------------|------------|---------|-------|------|------|
> | Hand4Whole (ResNet-18)    | 49.46      | 8.55    | 97.3  | 50.9 | 35.9 |
> | BiDRN-h (Ours)            | 47.78      | 7.45    | 86.0  | 49.0 | 27.9 |
> | Hand4Whole (L1 pruning)   | 25.51      | 2.76    | 146.3 | 79.3 | 45.0 |
> | BiDRN (Ours)              | 17.22      | 2.50    | 118.3 | 70.8 | 37.6 |
>
> `Q2-3:` The qualitative results show that some meshes predicted by the proposed method do not align with images well (e.g., hands and faces in Figure 7).
>
> `A2-3:` Thank you for your feedback. We add more explanations below.
> 1. Considering that BiDRN uses only **22.1% parameters and 14.8% operations** of Hand4Whole, the prediction cannot match the full-precision method but is quite well. Compared to other BNNs, our BiDRN aligns the body quite better. Other BNNs might totally fail with unmatched human mesh.
> 2. As shown in Figure 7, we add a column for Hand4Whole. Interestingly, even the full-precision method occasionally struggles to align accurately with the images, as seen in **the second row of Figure 7**.
>
> `Q2-4:` It would be better to include the results of the baseline (Hand4whole) in the visual examples.
>
> `A2-4:` Thank you for your valuable suggestion. In the refined version, we have added the results of Hand4Whole in visual comparison.
>
> `Q2-5:` Why binarized deconvolution and linear layers even yield better performance (L345-346)?
>
> `A2-5:` Thank you for your question. Since BoxNet is designed to predict hand and face boxes, which require only a small number of parameters, we hypothesize that using a dense network in this context might be redundant and could potentially result in overfitting. We hope this observation proves valuable for future research.

---

> ### Author Response · Authors · 2024-11-22
> **Response to Reviewer kuYg (denoted as R2) part 2**
>
> `Q2-6:` How is the prediction head of SMPLX parameters binarized?
>
> `A2-6:` Thank you for your question. Following prior work in the binarization field (e.g., retaining full precision in the prediction head for large language models), we maintain the prediction head in full precision. This ensures that the output range will not be restricted by the effects of binarization.
>
> `Q2-7:` A visual architecture comparison with other binarized methods could help to capture the gist and novelty of the proposed design, especially when previous methods have similar components, e.g., Bi-real also adopts a piecewise polynomial function and shortcuts.
>
> `A2-7:` Thank you for your suggestion. We have added architectural comparisons in **Figure 2 and Figure 3 of the Appendix** to highlight the differences between BNN, Bi-Real, and our BiDRN. Specifically:
> 1. In the **Identity Layer**, BiDRN introduces RPReLU to redistribute the binarized output before applying the residual connection.
> 2. In the **Down Sample Layer (and other convolutional modules redesigned by BiDRN)**, Bi-Real cannot apply a residual connection due to the dimension mismatch issue. In contrast, BiDRN redesigns the Down Sample module to integrate the residual connection, preserving as much full-precision information as possible.
>
> `Q2-8:` Typos: "e.g.the" -> "e.g., the", "Hands regions" -> "Hand regions".
>
> `A2-8:` Thanks for pointing out the typos, we have refined them in the new version.

---

> > ### Comment · Reviewer_kuYg · 2024-11-27
> >
> > Thank the authors for the reply. Although the rebuttal has addressed a few of my concerns about the architecture design, I've decided to maintain my score for the following two reasons:
> > - I share the same concerns of Reviewers #fsnC and  #Ywt4 that the scope and the application of the proposed method are limited.
> > - The proposed method does not include cutting-edge competitors for this task and claims this is because they are proposed for SMPL but this paper aims at SMPLX. However, since SMPL is a subset of SMPLX (though the former cannot be converted to the latter directly), I don't understand why we cannot remove the head and hand prediction parts from the proposed network and train it for comparison.

---

> > > ### Author Response · Authors · 2024-11-28
> > >
> > > Thank you for taking the time to review our response. We add more explanations and hope that these can further address your concern.
> > >
> > > 1. We would like to clarify that our aim is not to create a general binarized framework that supports all types of HMR backbones, as this would be too expansive for a single study. By focusing on CNN-based backbones, we already cover a broad spectrum of methods, including the most recent approach, AiOS.
> > > 2. Thank you for pointing that out. In our previous response, we intended to highlight that a direct comparison is not fair, as our method targets SMPLX, while the competitors you mentioned are based on SMPL. That said, it is likely feasible to train a version by removing the head and hand prediction parts, as you suggested. However, this would require additional time and effort, and we plan to explore this approach later, including the comparison in the final version.

---

### Official Review · Reviewer_fsnC · 2024-11-04

**Soundness:** 3
**Presentation:** 3
**Contribution:** 2
**Rating:** 5
**Confidence:** 4

**Summary:**

This paper proposes a binarized network for the task of human shape recovery from images. The main design in the proposed Binarized
Dual Residual Network (BiDRN) is the Binarized Dual Residual Block (BiDRB), which is further composed of Local Convolution Residual (LCR) and Block Residual (BR). The LCR is extended to convolution operations with down scale, down sample, fusion up, and fusion down. The design of LCR includes using Hardtanh as pre-activation function and the adoption of RPReLU activation. Experiments are conducted on two datasets to show the effectiveness of the proposed method compared with other binarization methods.

**Strengths:**

This paper presents detailed design of the binarized dual residual block, which covers the design of the pre-activation function, local convolution residual with channel-wise RPReLU, and block residual.

The Local convolution residual is extended to scenarios where the residual does not match the dimension of the outputs of convolutions.

Both the backbone of the reconstruction task and the face and hand detection network are considered in the network binarization.

Experiments on EHF and AGORA show performance advantages over other binarization methods.

**Weaknesses:**

The novelty and contribution of the proposed binarization is limited as most of the design is quite straightforward. The block residual is something new to me.

The designed residual block is quite specific for ResNet. This limits the scope of the proposed binarized dual residual block.

The experiment is conducted with comparison to other standard binarization methods, which are designed for general networks. This comparison is kind of unfair as the proposed binarization only works for this particular model, or maybe a broader ResNet-like architecture. A comparison to other binarization/quantization methods for networks in 3D human reconstruction or ResNet-like models is needed.

**Questions:**

It is not very clear why the proposed binarization method target at the Hand4Whole method. There are other (latest) methods (with better performance) in the field.

Latency. While the OPs is provided to have a rough idea of the number of operations, It is also necessary to see the change of speed with the proposed binarization.

---

> ### Author Response · Authors · 2024-11-22
> **Response to Reviewer fsnC (denoted as R1)**
>
> `Q1-1:` The novelty and contribution of the proposed binarization is limited as most of the design is quite straightforward. The block residual is something new to me.
>
> `A1-1:` Thank you for your feedback. We hope that our BiDRN is easy to follow with these straightforward designs. Experiments show that they are quite effective and provide a large improvement compared to other BNNs. Guided by simple and effective principles, our BiDRN could serve as a foundation of binarized HMR.
>
> `Q1-2:` The designed residual block is quite specific for ResNet. This limits the scope of the proposed binarized dual residual block.
>
> `A1-2:` Thank you for your feedback. Although the residual block replaces the ResNet in main experiments, it can be applied to broader **CNN-based** backbones. For example, we have also migrated the BiDRB to MobileNet, as shown in **Table 6 of the Appendix**.
>
> | **Method**                          | **Params (M)** | **OPs (M)** | **MPJPE$\downarrow$**  |
> |---------------------------------|------------|---------|---------|
> | MobileNetv1 (full-precision)    | 3.2        | 583.3   | 176.5   |
> | BNN                              | 0.2        | 17.9    | 338.3   |
> | BiDRN (Ours)                    | 0.2        | 17.9    | 188.3   |
>
> We binarized MobileNet using BNN and our proposed BiDRN, and evaluated the models on a 2D keypoint prediction task. The results highlight the effectiveness and versatility of our BiDRN approach.
>
>
> `Q1-3:` The experiment is conducted with comparison to other standard binarization methods, which are designed for general networks. This comparison is kind of unfair as the proposed binarization only works for this particular model, or maybe a broader ResNet-like architecture. A comparison to other binarization/quantization methods for networks in 3D human reconstruction or ResNet-like models is needed.
>
> `A1-3:` Thank you for your suggestion. We add more explanations and experiments below.
> 1. The compared BNNs are actually designed for CNN, and most of them are applied to ResNet only in their experiments.
> 2. Our BiDRN is not restricted to ResNet-like architecture, it can be adapted to any **CNN** backbone. Considering that the compared BNNs are proposed for CNNs, we consider it a fair comparison.
> 3. To the best of our knowledge, BiDRN is the first work for binarized 3D human reconstruction and no other binarization methods target this region. Yet, we also follow your suggestion to **compare with one more binarization method BBCU**. Although it is not designed for 3D human reconstruction, it is also applied in ResNet-like models (SRResnet).
>
> | **Method**                          | **Params (M)** | **OPs (G)** | **MPVPE (All)$\downarrow$**  | **MPVPE (Hand)$\downarrow$**  | **MPVPE (Face)$\downarrow$**  |
> | ------------------------------- | ---------- | ------- | ------- | ------- | ------- |
> | BBCU                            | 22.04      | 2.68    | 164.9   | 89.7 | 50.3 |
> | BiDRN (Ours)                    | 17.22        | 2.50    | 118.3   | 70.8 | 37.6 |
>
> The results show that our BiDRN also largely outperforms BBCU with fewer parameters and operations.
>
> `Q1-4:` It is not very clear why the proposed binarization method target at the Hand4Whole method. There are other (latest) methods (with better performance) in the field.
>
> `A1-4:` Thank you for your feedback. At the beginning of this project, Hand4Whole is the SOTA method. Yet, our BiDRN can be adapted to other latest methods. We added an experiment by adapting our BiDRN to the **latest method MultiHMR [ref1]**. Due to the time and resource limit, we haven't finished the experiments. Yet, after training for the same number of iterations and evaluation on EHF, our BiDRN largely surpasses BNN, showing that it can be generalized to other methods in the field.
>
> | **Method**                          | **\#Iteration** | **EHF** | | **Bedlam** | |
> |---------------------------------|---------|---------|---------|---------|---------|
> | | |**PVE$\downarrow$**  | **PAPVE$\downarrow$**  |**PVE$\downarrow$**  | **PAPVE$\downarrow$**  |
> | BNN                             | 70,000        | 344.9 | 168.2 | 292.4 | 151.1 |
> | BiDRN (Ours)                    | 70,000        | 302.7 | 137.1 | 270.2 | 126.5 |
>
> [ref1] Multi-HMR: Multi-Person Whole-Body Human Mesh Recovery in a Single Shot, ECCV, 2024.
>
> `Q1-5:` Latency. While the OPs is provided to have a rough idea of the number of operations, It is also necessary to see the change of speed with the proposed binarization.
>
> `A1-5:` Thank you for your suggestion. We have provided the latency comparison in **Table 7 of the Appendix**, where our BiDRN achieves impressive acceleration compared to the full-precision method.

---

> > ### Comment · Reviewer_fsnC · 2024-11-25
> > **Thanks for the response**
> >
> > Q1-2: The results in Table 6 can not be used to show the effectiveness of the proposed method for 3D Whole-body Human Mesh Recovery, which is the focus of this paper. I will advise the authors to adjust the scope of this paper if they think the results in Table 6 would be useful. This could be general tasks in human keypoint detection (as in Table 6) and human mesh recovery or beyond. But that will need significant change to all sections of the paper, which should be a new paper.
> >
> > Q1-3: If the proposed method could be extended to any CNN-like models, the contribution would be the introduction of RPReLu in Eq. 6 as identified in the Figures 2&3 in the supplementary file. This will leads to two questions. 1) Why other types of networks are not tested in the experiments to verify this? 2) The contribution seems to be a general modification on a backbone, which does not have any direct link with the human mesh recovery task. Then, why the authors propose the RPReLu for human mesh recovery only? What is the motivation? Why not other tasks as well?
> >
> > Q1-4: The focus of the Hand4Whole method does not only exist in the experiment. It also lies in the methodology part.
> >
> > Q1-5: There is no explanation about what do the numbers in the Table 7 of the supplementary file mean.
> >
> > Since my concerns are not fully addressed, I will maintain my rating at the moment.
> > I share the same concerns as Reviewer Ywt4 and Reviewer kuYg on the design and experiment in this paper.

---

> > > ### Author Response · Authors · 2024-11-28
> > >
> > > Thank you for taking the time to review our response. We add more explanations below.
> > >
> > > `A1-2:` Thank you for your suggestion. We agree that Table 6 may not fully address the concern regarding our method being limited to ResNet in the HMR task. However, we believe the results for `A1-4` provide evidence that our method can also be adapted to other convolutional networks, such as ConvMixer, as demonstrated in the HMR task experiments.
> > >
> > > `A1-3:` We would like to clarify that our contributions extend beyond the RPReLU and include the following:
> > > - Redesigned modules to bypass full-precision activations during dimensional changes, addressing a challenge not tackled in previous work.
> > > - The Block Residual (BR) module, illustrated in Figure 4, which you have also acknowledged as novel.
> > > - The binarization of BoxNet, specifically tailored for the HMR task, to predict hand and face boxes effectively.
> > >
> > > In addition, we conducted further experiments using the ConvMixer backbone in `A1-4` to demonstrate the generalizability of our approach to other types of networks.
> > >
> > > `A1-4:` As explained in our previous response, Hand4Whole was the state-of-the-art method at the time we began this work. Since Hand4Whole serves as a fairly general framework, we chose to base our description on it to facilitate better understanding.
> > >
> > > `A1-5:` Thanks for pointing it out. The latency shown in the table is measured based on processing a single image with dimensions 3×224×224, where the image has 3 channels and a spatial resolution of 224×224. We will add it in the final version.
> > >
> > > We hope that these explanations can further address your concerns.

---

> > > > ### Comment · Reviewer_fsnC · 2024-11-28
> > > > **Thanks for the response**
> > > >
> > > > A1-2: The concern remains as the proposed method is now limited to CNN-like methods.
> > > >
> > > > A1-3: As pointed out in my previous response, my concerns are not addressed even though the contributions are extended to include the module redesign to bypass full-precision activations during dimensional changes, Block residual module.,
> > > >
> > > > A1-4: My concern remains as there is no description in the methodology part on how the proposed method could be used for other types of models or HMR models. This will require significant changes to the paper although this is something could be foreseen. It can not be guaranteed that such changes could be included in the final version. I think that would need another round of review for the updated content.
> > > >
> > > > A1-5: It is still not clear to me what ‘the latency’ means? Further explanation of the numbers is expected.
> > > >
> > > > Though this paper could be promising with significant changes, I think that would need another round of review. I will maintain my rating as negative.

---

### Author Response · Authors · 2024-11-22
**Response to all reviewers and area chairs for a brief summary**

Dear reviewers and area chairs,

We sincerely thank all reviewers and area chairs for their valuable time and insightful comments.

We are pleased to note that:
1. Reviewers kuYg and zufd find our method easy to follow, with reviewer zufd also recognizing the solidity of our motivation.
2. Reviewer Ywt4 acknowledges that our method effectively addresses an underexplored challenge in the binarized HMR task.
3. All reviewers appreciate the superior performance of our method compared to previous BNN approaches.
4. Reviewers Ywt4 and zufd highlight that our method achieves significant reductions in both parameters and computational operations.

We have responded to each reviewer individually and would like to summarize our responses here:
1. We clarify that our method is applicable to a wider range of **CNN-based** backbones.
2. We perform an additional experiment to **compare our BiDRN with the more recent binarization method BBCU**.
3. We adapt our method to the **latest one-stage HMR method, MultiHMR**, further showcasing its generalizability and effectiveness.
4. We include the **visual comparison of the full-precision method Hand4Whole**.
5. We include a **visual architecture comparison** between our BiDRN and other BNN methods to better illustrate our novelty.
6. We clarify details in the paper, including the **latency improvement**, **design of down sample residual block**, **comparison to efficient methods**, **qualitative results**, and **motivation of binarizing BoxNet**.

We extend our gratitude again to all reviewers and area chairs!

Best regards,

Authors

---

### Meta-Review · Area_Chair_w1xX · 2024-12-21

**Metareview:**

This paper proposes a binarized network for whole-body 3D human mesh recovery. The proposed method is designed to reduce the high computational cost of existing multi-stage pipeline, so that the model can efficiently run on mobile devices. Built on the baseline, Hand4Whole, this paper redesigns the architecture of backbone, bounding box detector, and decoding heads to reduce the computational costs while alleviating the reduction in precision.
Although the performance gain by the proposed method over existing methods should be appreciated, the reviewers feels that the paper provides an engineering solution to the task.  Indeed, the reviewers except for Reviewer zufd raised common concerns regarding limited novelty, insufficient validation, and lacking detailed explanation of the method.  On the other hand, Reviewer zufd appreciates the first implementation of binarized network for this task and good performance.  The architecture is straightforward and the design choices are not well justified.  Moreover, it is specific to CNN-based backbone.  Compatibility with one-stage methods is unclear.  It is also suggested to choose appropriate methods for HMR to compare.  The authors tried to address the raised concerns by providing additional experiments; however, the reviewers except for Reviewer zufd have not been convinced.  In fact, newly provided experiment on human keypoint detection is not in the focus of this paper, meaning significant changes of the scenario will be required.  Reviewer kuYg’s concern on missing comparison with methods for SMPL is reasonable.  Even though the aim is not to create a general binarized framework, comparison is necessary to convince the effectiveness of the proposed method.  The authors say that the Local Convolution Residual and Block Residual techniques are compatible with one-stage methods, but no evidence is provided.  It is unconvincing to focus on the model with CNN backbone as there are more advanced methods with ViT backbones.  Simpleness and effectiveness cannot be significant contribution.  Clarifying the design ideas behind should be clearly explained with evidence to bring some insight to the community.  Otherwise, such a paper cannot be solid.  The claims of the paper should be properly justified with thorough evaluation.  For this, substantial improvements and another round of reviews are required. This paper cannot be accepted, accordingly.

**Additional Comments On Reviewer Discussion:**

See above.

---

### Decision · Program_Chairs · 2025-01-22

Reject